# Uncertainty-Aware Safety Propagation Critics for Safe Reinforcement Learning

**Kutay Demiray**$^*$                    *kutay.demiray@bilkent.edu.tr*
*Department of Computer Engineering*
*Bilkent University*

**Efe Eren Ceyani**$^*$                    *eren.ceyani@bilkent.edu.tr*
*Department of Electrical and Electronics Engineering*
*Bilkent University*

**Özgür Salih Öğüz**                    *ozgur@cs.bilkent.edu.tr*
*Department of Computer Engineering*
*Bilkent University*

**Reviewed on OpenReview:** *https://openreview.net/forum?id=b8GWqgLEHh*

## Abstract

Safe reinforcement learning (RL) aims to optimize long-term performance while satisfying safety constraints, a requirement that is critical in many applications but difficult to guarantee when cost estimates are inaccurate or data is limited. In model-free actor-critic methods, cost critics are often unreliable in poorly explored regions, leading to constraint violations during both training and deployment. In this work, we propose a novel uncertainty-aware approach in safe RL called USPC, which constructs conservative cost surrogates using epistemic uncertainty. Our method trains an ensemble of cost critics to estimate uncertainty and uses these estimates to build an upper confidence bound on predicted costs. We then introduce a safe set network that approximates a pessimistic surrogate of the cost action-value function inspired by safe Bayesian optimization, enabling scalable safety propagation in continuous state-action spaces. Replacing standard cost critics with this surrogate in existing off-policy safe RL algorithms yields policies that are less likely to violate cost constraints. We show empirically across multiple Safety Gymnasium benchmark tasks that our approach reduces both the frequency and magnitude of constraint violations in most tasks while maintaining competitive reward performance compared to the baselines. We further provide theoretical justification for our approach, including a conservativeness guarantee for the safe set network, and an approximation result for the anchor-based scheme. Project website is available at `https://sites.google.com/view/uspcrl`.

## 1    Introduction

Reinforcement learning (RL) has shown promise in solving complex tasks in diverse domains, including robotic control (Kober et al., 2013), games (Mnih et al., 2015), and natural language processing (Ouyang et al., 2022). However, most of RL's achievements so far have been in simpler simulated or digital environments (Dulac-Arnold et al., 2021), while RL in real-world tasks, such as autonomous driving with a real car (Zhang et al., 2024), faces several obstacles. One major obstacle is that reinforcement learning policies often violate some desired safety constraints (such as an autonomous car with the goal of arriving at a target point should also avoid collisions), which makes collecting real-world data expensive or even outright impossible.

---

$^*$Equal contribution

To address this issue, safe reinforcement learning deals with algorithms that learn a cumulative reward maximizing policy while also constraining cumulative costs to be below some predefined threshold. However, learning such a policy faces some further difficulties, especially when black box models such as deep neural networks are used. If the safety constraint is active (i.e., actually constraining the optimal solution), the gradients for reward maximization and cost reduction are often conflicting, which makes learning difficult. Furthermore, without the use of an explicit world model, the predicted future cumulative cost can be inaccurate, and thus constraint satisfaction cannot be guaranteed.

Existing works in safe learning-based control use uncertainty, particularly epistemic uncertainty, in various ways to account for poorly explored regions of the data. Early approaches to safety under uncertainty were explored in the context of safe Bayesian optimization, which leveraged Gaussian process uncertainty estimates to ensure constraint satisfaction while exploring unknown regions (Sui et al., 2015). However, these methods were primarily developed for discrete or low-dimensional settings, limiting their applicability to the continuous state and action spaces, which are typical in RL. Some recent model-based methods have used the uncertainty of the world model to facilitate safer trajectories, for example, by applying it as an intrinsic penalty for the model predictive control planner in a soft-constrained setting (Lütjens et al., 2019). However, careful tuning is necessary with such algorithms; since the model has no explicit definition of safety, it is possible for the algorithm to yield trajectories that yield seemingly good objective values but would be considered unsafe from a human perspective. Other model-based works have proposed algorithms that use a Gaussian process world model with a reachability-based approach to satisfy hard constraints (As et al., 2025). While such methods often provide strong theoretical guarantees, they also tend to extend poorly to continuous state or action spaces, or large amounts of data. To address this issue, model-free safe RL methods have also received significant attention. In particular, since safety is often defined in terms of cost return in a constrained Markov decision process (CMDP), many methods leverage the epistemic uncertainty in value (or value-like) function estimators to improve safety (Stachowicz & Levine, 2024; Günster et al., 2024).

In this work, we address the problem of unreliable cost critics in deep model-free safe RL by introducing a safety propagation mechanism inspired by safe Bayesian optimization. Safe BO methods such as SafeOpt (Sui et al., 2015) and its extensions (Berkenkamp et al., 2023) provide a principled framework for propagating safety information between nearby decisions under Lipschitz regularity assumptions, but they are fundamentally tied to Gaussian process surrogates and discrete or low-dimensional decision spaces. Meanwhile, existing model-free safe RL methods that leverage epistemic uncertainty typically treat each state-action pair independently, constructing pointwise conservative cost estimates without propagating safety information across the action space. We propose Uncertainty-Aware Safety Propagation Critics (USPC), a method that aims to bridge this gap: it replaces GP confidence bounds with neural network ensemble-based uncertainty and replaces discrete safe set enumeration with a learned safe set network (SSN), enabling continuous-space safety propagation within an actor-critic framework. The resulting cost surrogate is typically more conservative than the ensemble mean in high-uncertainty regions while remaining compatible with standard off-policy safe RL training. Our contributions can be summarized as follows:

- We propose USPC, the first method to extend the safety propagation principles of safe Bayesian optimization to safe model-free RL with continuous state-action spaces using neural network function approximation. By replacing Gaussian process confidence bounds with ensemble-based uncertainty estimation and learning a safe set network as a continuous surrogate for the discrete safe set, USPC enables continuous action space safety propagation in deep RL without relying on GPs or finite action sets.

- We show that USPC functions as a modular, plug-and-play component for existing off-policy safe RL algorithms. Replacing the cost critic with the learned SSN in TD3-Lag and CVPO yields policies with lower constraint violation rates for most tasks while maintaining competitive reward performance.

- We provide formal justification for the safety propagation mechanism, including a conservativeness guarantee for the SSN relative to the ensemble mean, a high-probability bound showing that the

SSN upper-bounds the true cost Q-value, and approximation results for the anchor-based scheme, alongside a discussion of the design choices, assumptions, and limitations of the approach.

We demonstrate our contributions with experiments on 6 Safety-Gymnasium tasks. Empirically, we show that USPC yields similar reward curves while keeping costs below constraints more consistently. More specifically, we show that USPC's training episodes are safer compared to the baselines.

## 2 Related Work

**Safe Bayesian optimization.** Optimization and exploration under uncertainty has been studied for a long time in the bandit literature (Bubeck et al., 2012). These works were then used in settings where reward functions could be modeled by a Gaussian process, both enabling the usage of confidence bounds and establishing a connection to Bayesian optimization (Srinivas et al., 2009). Safe Bayesian optimization extends this by enforcing safety constraints throughout learning via uncertainty-aware confidence bounds and regularity assumptions. Sui et al. (2015) used confidence bounds and regularity assumptions to maintain a certified set of safe actions and expand it through safety propagation with probabilistic guarantees. Following works have extended this work to multiple safety constraints (Berkenkamp et al., 2023) and stagewise optimization strategies that decouple safety propagation from utility maximization (Sui et al., 2018; Wachi & Sui, 2020). Similar Gaussian process based safety propagation ideas have also been used in other areas such as safe exploration and control settings (Turchetta et al., 2016; Berkenkamp et al., 2017).

While safe Bayesian optimization provides a principled framework for uncertainty-aware safety propagation with strong theoretical guarantees, it is not directly applicable to reinforcement learning immediately due to its reliance on Gaussian processes, discrete action sets, and low-dimensional settings. Our work draws inspiration from this line of work but replaces Gaussian process confidence bounds and explicit safe set construction with neural approximations that work over continuous state-action spaces. We embed the inspired safety propagation into an actor-critic framework by learning a pessimistic cost surrogate function.

**Safe model-free reinforcement learning.** Safe reinforcement learning has received significant attention following the success of unsafe reinforcement learning algorithms elsewhere. Often in these works, the MDP is replaced with a CMDP, where an episode is considered safe if the episodic cost return is below some threshold. A large proportion of modern safe RL focuses on primal-dual Lagrangian methods in a model-free setting. The simplest of these methods augment standard unsafe RL algorithms with a Lagrangian objective: In the primal step, they obtain the Lagrangian for the constrained optimization problem and maximize for rewards. Then, in the dual step the dual variables are updated. Related to this approach, RCPO (Tessler et al., 2018) proposes a multi-timescale approach where the dual variables are updated less frequently to stabilize training, and PID-Lagrangian (Stooke et al., 2020) extends the basic Lagrangian dual update with a PID controller to improve stability and reduce oscillations. Another seminal Lagrangian-based work is CPO (Achiam et al., 2017), which formulates each policy update as a constrained second-order trust-region optimization problem, ensuring that the new policy remains close (in KL-divergence) to the old one while keeping the expected cumulative constraint costs below specified limits. This design provides near-constraint satisfaction guarantees throughout training, but is computationally costly due to its second-order updates. Future CPO-based work has then improved on it in various ways such as PCPO decoupling reward optimization and constraint satisfaction with a projection step (Yang et al., 2020), and FOCOPS replacing the second-order updates with a more scalable first-order step (Zhang et al., 2020), and more recently C-TRPO augmenting the trust region update so that it only contains safe policies (Milosevic et al., 2025).

However, these primal-dual methods often suffer from unstable learning dynamics and slow convergence due to coupled training of model parameters and dual variables, hence a variety of primal-only methods have been proposed to avoid those drawbacks. CRPO (Xu et al., 2021) proposes a mechanism to switch gradients between reward improvement when the policy is safe and cost reduction when the policy is unsafe, which PCRPO (Gu et al., 2024) then expands to reduce performance degradations and oscillations near the constraint boundary with a projection approach. Later works (Wu et al., 2024) have augmented the primal-dual approach by introducing pessimism via critic ensembles, similar to our work. IPO (Liu et al.,

2020) transforms the constrained problem into an unconstrained one by adding a logarithmic barrier term to the primal objective.

A related line of work uses Hamilton-Jacobi reachability analysis to characterize feasible sets in model-free safe RL. RCRL (Yu et al., 2022) learns a safety value function capturing worst-case constraint violation to define the largest feasible set, though it is limited to deterministic settings. RESPO (Ganai et al., 2023) extends this to stochastic environments by learning a reachability estimation function predicting the probability of future violations. FISOR (Zheng et al., 2024) adapts this framework to the offline setting, identifying the largest feasible region from static data via reversed expectile regression and extracting policies through a feasibility-dependent diffusion model. Note that these methods propagate safety across states over time, characterizing which states can remain safe indefinitely, whereas our work considers an analogous propagation but across actions at a given state, certifying nearby actions as safe via Lipschitz continuity.

Another family of works uses epistemic uncertainty in safe reinforcement learning. A unique problem in safe RL compared to some other safe control methods is that since the underlying MDP is unknown, exploration is critical for obtaining a near-optimal policy, however the agent often lacks sufficient evidence to accurately estimate future costs, resulting in unsafe behaviors. Epistemic uncertainty can be used to provide a principled signal for this uncertainty, enabling safe RL algorithms to act conservatively and explore safely. One branch of these methods quantifies and uses the uncertainty in the world models. In online model-based safe RL, compounding model errors can make seemingly safe predicted trajectories unsafe in reality, hence the epistemic uncertainty can be used to account for this unreliability (Berkenkamp et al., 2017; Wachi & Sui, 2020; As et al., 2025). In offline model-based settings, epistemic uncertainty can additionally help with preventing unsafe behaviors in regions insufficiently covered by the offline data (Yu et al., 2020; Kidambi et al., 2020; Diehl et al., 2023). However, all of these methods can still suffer from model inaccuracy in cases where the environment is too complex to learn dynamics accurately without prohibitively large datasets or additional training, such as high-dimensional, contact-rich or discontinuous environments. In existing model-free safe RL works, uncertainty has been used in diverse ways, including conservative constraint enforcement (Bharadhwaj et al., 2021; Zhang et al., 2024; Stachowicz & Levine, 2024), dual variable tuning in primal-dual methods (Zhang et al., 2024), and safety filters (Eilers et al., 2023; Flögel et al., 2025). RACER (Stachowicz & Levine, 2024) combines ensemble-based distributional critics with a CVaR objective and adaptive action limits to reduce failures during training, though their method optimizes a risk-sensitive reward return rather than enforcing cost constraints in a CMDP. In a related but partially model-based direction, D-ATACOM (Günster et al., 2024) learns a distributional feasibility value function with CVaR-based risk-aware constraints within the ATACOM framework (Liu et al., 2022a), combining model-free policy learning with a known dynamics model for constraint enforcement, and targeting stepwise rather than cumulative cost constraints.

Other approaches to safe RL include Lyapunov-based methods (Chow et al., 2018; 2019), action projection and shielding (Dalal et al., 2018; Wagener et al., 2021), and safe set/reachability-based methods using model-free approximations (Wachi & Sui, 2020). Barrier certificate-based methods (Zhao et al., 2022; Ma et al., 2025) introduce a barrier function to model-free RL algorithms that aims to establish forward invariance for the safe region/set and bring constraint violations to zero or near-zero.

In contrast to these methods, USPC introduces a learned safety surrogate that directly replaces the cost critic and propagates safety information across the action space. This allows safety information obtained at a limited set of actions to generalize to nearby actions, reducing constraint violations caused by sparse or inaccurate cost estimates. As a result, our method preserves the simplicity and sample efficiency of model-free actor–critic algorithms while addressing a core limitation shared by many existing approaches, which enforce safety only at sampled actions or along the current policy trajectory, without explicitly propagating safety to nearby actions in the continuous action space.

**Epistemic uncertainty-based methods in deep reinforcement learning.** Epistemic uncertainty is central to a wide variety of problems in reinforcement learning, apart from safety. While most reinforcement learning methods only implicitly reduce the epistemic uncertainty, a significant amount of works explicitly quantify and use epistemic uncertainty to address various problems.

Popularly, epistemic uncertainty-based methods use the uncertainty to guide exploration. One way to use the uncertainty is to construct an intrinsic reward that scales with uncertainty, so that the policy will be more likely to visit these underexplored regions. The intrinsic reward can be constructed in various ways, such as using visitation counts (Auer et al., 2002; 2008), disagreement in ensemble transition models (Shyam et al., 2019; Sekar et al., 2020), information gain (Sun et al., 2011; Houthooft et al., 2017). Some other works utilize epistemic uncertainty to do posterior sampling of value functions (Osband et al., 2016; 2018; Azizzadenesheli & Anandkumar, 2019; An et al., 2021) or to obtain upper confidence bound-based optimistic value function (Lee et al., 2021; Schmitt et al., 2023) to guide exploration.

Our work is a model-free safe RL method utilizing epistemic uncertainty, but we use a Lipschitz safety propagation-based statistic similar to (Sui et al., 2015). To the best of our knowledge, existing model-free works often utilize uncertainty using simpler statistics such as the upper confidence bound, whereas a safety propagation/reachability-based RL approach has only been applied in model-based settings to quantify uncertainty in learned dynamics (Wachi & Sui, 2020; As et al., 2025).

**Distributional reinforcement learning.** A complementary line of work addresses safety through distributional reinforcement learning, which models the full distribution of returns rather than point estimates. Building on the distributional RL foundations of Bellemare et al. (Bellemare et al., 2017) and quantile regression methods (Dabney et al., 2018a;b), several works have applied distributional critics directly to cost estimation in constrained MDPs. WCSAC (Yang et al., 2021; 2023) learns a distributional safety critic using quantile regression and constrains the CVaR of cumulative costs, showing that tail-risk constraints yield safer policies than expected-cost constraints in stochastic environments. SDAC (Kim et al., 2023) extends this with trust-region updates for multiple constraints, while SRCPO (Kim et al., 2024) generalizes to spectral risk.

A key distinction from our work is that distributional methods primarily capture aleatoric uncertainty, whereas our ensemble-based approach targets epistemic uncertainty arising from limited data or poor exploration. Some works combine both paradigms: SENTINEL (Eriksson et al., 2022) proposes a composite risk framework applying aleatoric risk measures within each ensemble member and epistemic risk measures across the ensemble. More recently, COX-Q (Li et al., 2026) combines an ensemble of distributional cost critics with cost-constrained optimistic exploration, which we compare against empirically in Appendix C. USPC differs by combining epistemic uncertainty with Lipschitz-based safety propagation rather than tail-risk measures, allowing safety information to generalize across the action space.

## 3 Background

### 3.1 Problem Setting

Constrained Markov decision processes (CMDPs) are a common mathematical framework used for safe reinforcement learning. We define a CMDP by a tuple $(\mathcal{S}, \mathcal{A}, P, r, \gamma, \rho_0, C)$, where $\mathcal{S}$ and $\mathcal{A}$ denote state and action spaces, $P : \mathcal{S} \times \mathcal{A} \times \mathcal{S} \rightarrow [0, 1]$ is the transition kernel that contains probabilities $P(s_{t+1}|s_t, a_t)$, $r : \mathcal{S} \times \mathcal{A} \rightarrow \mathbb{R}$ is the reward function, $\gamma \in [0, 1]$ is the discount factor, $\rho_0 \in \mathcal{P}(\mathcal{S})$ is the probability distribution of initial states, and $C = \{c_i : \mathcal{S} \times \mathcal{A} \rightarrow [0, c_{max}], i = 1, \ldots, |C|\}$ is the set of cost functions that define the safety related constraints. We denote possible minimum and maximum action values as $a_{\min}$ and $a_{\max}$, respectively. A trajectory is a sequence $(s_t, a_t)_{t=0}^{T}$, where $t$ denotes a timestep and $T$ denotes the horizon (possibly infinite). At every timestep $t$, the agent observes the state $s_t \in \mathcal{S}$, samples an action $a_t \sim \pi(\cdot|s_t)$, and the environment transitions to $s_{t+1} \sim P(\cdot \mid s_t, a_t)$.

For any stationary policy $\pi$, the reward action-value function and the cost action-value function for constraint $i$ are defined as

$$Q_r^\pi(s, a) = \mathbb{E}_\pi \left[ \sum_{t=0}^{\infty} \gamma^t \, r(s_t, a_t) \, \middle| \, s_0 = s, \, a_0 = a \right] , \tag{1}$$

$$Q_{c_i}^\pi(s, a) = \mathbb{E}_\pi \left[ \sum_{t=0}^{\infty} \gamma^t \, c_i(s_t, a_t) \, \middle| \, s_0 = s, \, a_0 = a \right] , \tag{2}$$

where the expectation is taken over trajectories induced by $\pi$ and $P$. In this paper, we consider the action-values for the current learned policy by default and omit the $\pi$ for notational simplicity.

Given a vector of cost thresholds $\boldsymbol{h} := [h_1, \ldots, h_M]^\top \in \mathbb{R}_{\geq 0}^M$, the goal of safe RL is to obtain a safe and optimal policy $\pi^*$ that maximizes expected reward while satisfying cost constraints:

$$\pi^* \in \arg\max_\pi \ J_r(\pi) := \mathbb{E}_{s_0 \sim \rho_0, \, a_0 \sim \pi(\cdot|s_0)}\big[Q_r(s_0, a_0)\big] \tag{3}$$

$$\text{s.t.} \ J_{c_i}(\pi) := \mathbb{E}_{s_0 \sim \rho_0, \, a_0 \sim \pi(\cdot|s_0)}\big[Q_{c_i}(s_0, a_0)\big] \leq h_i, \quad i = 1, \ldots, |C| \ . \tag{4}$$

In this work, we focus on a single safety constraint (i.e., $|C| = 1$) for clarity, and denote its cost function by $c$ and its threshold by $h$. Our method can be extended easily to multiple constraints, given that the base safe RL algorithm is compatible.

## 3.2   Actor-Critic Safe Reinforcement Learning

In actor-critic-based safe RL algorithms, the value functions defined in the previous subsection are represented using neural networks and optimized from replay data. The reward critic $Q_r(s, a)$ and cost critic $Q_c(s, a)$ are trained using temporal-difference (TD) regression on single-step targets:

$$y_t^r = r(s_t, a_t) + \gamma \mathbb{E}_{a' \sim \pi_\theta(\cdot|s')}\big[Q_r(s', a')\big] \ , \quad y_t^c = c(s_t, a_t) + \gamma \mathbb{E}_{a' \sim \pi_\theta(\cdot|s')}\big[Q_c(s', a')\big] \ . \tag{5}$$

The policy is then updated using a constrained objective based on the critics. A common approach is to form a Lagrangian where a multiplier $\lambda \geq 0$ penalizes unsafe behavior:

$$\mathcal{L}(\theta, \lambda) = \mathbb{E}_{s, a \sim \pi_\theta}\big[ -Q_r(s, a) + \lambda(Q_c(s, a) - h)\big] \ . \tag{6}$$

This results in alternating gradient updates on $\theta$ and dual updates on $\lambda$, and covers many primal-dual safe RL algorithms. Variants include penalty methods, trust-region formulations, and maximum-entropy actor-critic objectives.

Throughout this work, we follow this standard deep actor-critic paradigm, where the policy is represented by a neural network $\pi_\theta(a|s)$, and the action-value functions $Q_r$ and $Q_c$ are learned by minimizing TD losses over replayed transitions. This forms the basis for our deep safe learning setting.

## 3.3   Safe Bayesian Optimization

Safe Bayesian optimization provides a framework for decision making under uncertainty when function evaluations are expensive and safety constraints must be respected throughout the learning process. Rather than treating uncertainty only for exploration, safe Bayesian optimization also leverages epistemic uncertainty to prevent unsafe evaluations while gradually expanding the set of admissible decisions.

The main technique used in safe Bayesian optimization is the construction of uncertainty-aware confidence bounds. Given noisy observations of an unknown quantity, a Gaussian process with posterior mean $\mu_t(x)$ and variance $\sigma_t^2(x)$ is used. Then, uncertainty estimates are constructed using confidence intervals:

$$l_t(x) = \mu_t(x) - \beta_t^{1/2}\sigma_t(x), \quad u_t(x) = \mu_t(x) + \beta_t^{1/2}\sigma_t(x) \ , \tag{7}$$

where $\beta_t$ is chosen to control the level of conservativeness. In Gaussian process works, $\beta_t$ is typically chosen as a function of time to ensure that the confidence intervals contain the true function with high probability, which enables formal safety and regret guarantees under standard regularity assumptions (Srinivas et al., 2009; Sui et al., 2015; Rothfuss et al., 2023).

Sui et al. (2015) leverages this construction to define safety in a reachability-based manner. Rather than certifying safety only at individual decisions, they maintain a safe set that is iteratively expanded over time. At iteration $t$, a decision is considered safe if its upper confidence bound satisfies the safety constraint, or if it can be certified as safe by propagating safety from previously safe decisions using a regularity assumption.

Assuming Lipschitz continuity with $L$ with respect to some distance metric $d$, safety is propagated to decision $x$ if:

$$\min\{u_t(x), u_t(x') + Ld(x - x')\} \leq h \ . \tag{8}$$

This allows us to conservatively infer safety at unobserved points that are sufficiently close to the known safe ones.

The use of Gaussian process models enables principled uncertainty quantification and high-probability guarantees. However, this approach relies on assumptions that limit scalability, including the need to evaluate confidence bounds over discrete or low-dimensional domains and the cubic computational cost of Gaussian process inference. These limitations make direct application of these methods infeasible in reinforcement learning settings with continuous state and action spaces.

Nevertheless, the mathematical structure underlying this line of work is relevant to safe reinforcement learning. The combination of uncertainty-aware upper confidence bounds and Lipschitz-based safety propagation motivates the construction of conservative surrogates for cost evaluation. Our method adopts this structure while replacing Gaussian processes and explicit safe set representations with scalable neural approximations that are compatible with reinforcement learning.

## 4   Uncertainty-Aware Safety Propagation Critics

In this section, we present USPC, **U**ncertainty-Aware **S**afety **P**ropagation **C**ritics for Safe Reinforcement Learning, which combines ensemble-based cost estimation with a scalable safety propagation mechanism that can be seamlessly integrated into off-policy actor-critic algorithms. The overall USPC mechanism is illustrated in Figure 1 on a simple one-dimensional example. A high-level overview of safe RL training with USPC can be seen in Figure 2.

### 4.1   Ensemble Critics

Standard actor-critic RL methods train a single cost critic network $Q_c$. Instead, we train an ensemble of $M$ critics, denoted $\{\hat{f}_{\theta_i}\}_{i=1}^M$, where each member $\hat{f}_{\theta_i} : \mathcal{S} \times \mathcal{A} \to \mathbb{R}$ is initialized independently and trained through the same stochastic gradient updates using the same replay batches. Intuitively, the ensemble captures epistemic uncertainty arising from limited data or poorly explored regions, which allows us to construct confidence intervals similar to those in Gaussian process based safe Bayesian optimization. To construct meaningful confidence intervals, we treat the outputs of the ensemble members as random samples from an underlying distribution, and we more formally justify this view under mild assumptions:

**Assumption 1.** *Let $\hat{f}$ be an ensemble consisting of members $\hat{f}_{\theta_i} : \mathcal{S} \times \mathcal{A} \to \mathbb{R}$ for $i = 1, \ldots, M$ trained to fit a target function $f$, where the parameters $\theta_i$ are initialized independently and are identically distributed. Suppose that all $\hat{f}_{\theta_i}$ are trained in parallel using the same data batches presented in the same order. For a given $(s, a)$, let $\hat{y}_i(s, a) = \hat{f}_{\theta_i}(s, a)$ denote the member outputs, and define their sample mean and variance as $\hat{\mu}(s, a)$ and $\hat{\sigma}^2(s, a)$, respectively. Then, conditional on the training process (i.e. on the fixed data and shared batch sequence, such that the only randomness comes from the initialization of weights), the outputs $\hat{y}_i(s, a)$ are i.i.d. random variables.*

In regions of the input space that are sufficiently covered by data, ensemble members can be treated as approximately unbiased estimators of the true function $f$. This, alongside Assumption 1, implies that by the Central Limit Theorem, for sufficiently large $M$, the sampling distribution of $\hat{\mu}(s, a)$ more specifically converges to

$$\hat{f}(s, a) := \hat{\mu}(s, a) \sim \mathcal{N}\left(f(s, a), \frac{\hat{\sigma}^2(s, a)}{M}\right). \tag{9}$$

Assumption 1 follows from the fact that each ensemble member is independently initialized using the same prior distribution for $\theta_i$ and trained through the same deterministic procedure. Conditional on the shared training data and batch order, the outputs are i.i.d., and the Central Limit Theorem implies that their empirical mean distribution approaches a Gaussian as the ensemble size $M$ grows.

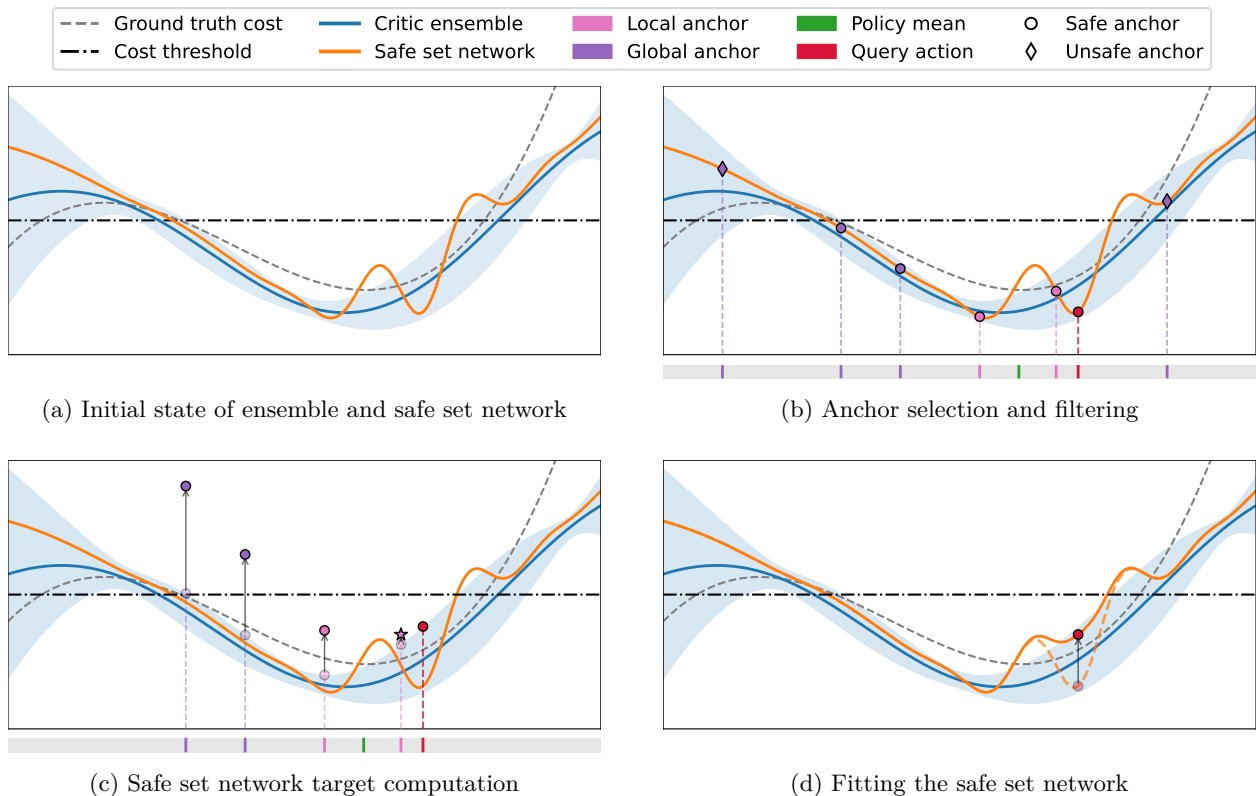

Figure 1: An illustration of one step of USPC training on a simple one-dimensional example. (a) Initial state of the critic ensemble and safe set network. Shaded blue region shows the confidence interval for the ensemble. (b) A query point (for safe set network update) is selected, and local and global anchors are sampled to create the anchor set. Anchor points are then filtered according to the current safe set network and the cost threshold, and unsafe points are discarded. (c) Safe anchor points are evaluated on the UCB, then shifted upwards by the Lipschitz distance term to obtain candidate targets. The minimum of the candidates (denoted by star) is selected as the target for the safe set network update. (d) Safe set network is updated at the query point via regression towards the target selected in the previous step.

Using ensemble mean and variance statistics, we construct an upper confidence bound (UCB) on the cost action-value function:

$$\text{UCB}(s,a) := \hat{\mu}(s,a) + \beta\hat{\sigma}(s,a) \ , \tag{10}$$

where $\beta \geq 0$ in our work is a hyperparameter controlling the level of conservativeness. We use this bound as a pessimistic surrogate for cost evaluation, increasing predicted costs in regions with high uncertainty.

Beyond providing a conservative pointwise estimate, the UCB is also Lipschitz, inheriting this property from the ensemble critics. This is useful in the analysis of the anchor-based SSN target, where we later bound SSN approximation error. We formalize this property below.

**Lemma 1.** *(Lipschitzness of UCB) Assume each ensemble member $\hat{f}_i$ is L-Lipschitz, and suppose that there exists an $\epsilon > 0$ such that $\hat{\sigma}_c(s,a) > \epsilon$ for any state-action pair $(s,a)$ [*]. Then, the upper confidence bound $UCB(s,a)$ is $L_{UCB}$-Lipschitz in actions, where*

$$L_{UCB} = L\left(1 + \frac{2\beta D}{\epsilon}\frac{M}{M-1}\right), \tag{11}$$

---

[*]In practice, this holds since ensemble members do not converge to the same point (due to local minima, etc.). Alternatively, one can explicitly enforce this condition by using $\tilde{\sigma}_c(s,a) := \sqrt{\sigma_c^2(s,a) + \epsilon^2}$ with a small $\epsilon$ value, instead of the regular definition of standard deviation.

*for* $D := \sup_u \max_i |\hat{f}_i(s,u) - \hat{Q}_c(s,u)|$.

The proof of Lemma 1 is available in Appendix A.3. A visualization of the UCB is also shown in Figure 1a (and the following subfigures).

## 4.2 Safe Set Network

### 4.2.1 Motivation

The critic ensemble $\hat{Q}_c$ that we train is not directly used to enforce safety. Instead, we train a safe set network (SSN) that approximates the safety of a state-action pair, denoted by $\hat{S}_c : \mathcal{S} \times \mathcal{A} \to \mathbb{R}$, as a substitute for the cost critics. SSN allows us to extend the safe set approach in reachability-based safe BO methods to continuous functions where the input space is infinite, while also scaling better than GPs used in these methods.

A natural question to ask is why the ensemble UCB itself cannot simply replace the cost critic. The UCB provides a pointwise pessimistic cost estimate: at each $(s,a)$ it inflates the predicted cost based on local ensemble disagreement. However, this treats every action independently. In continuous action spaces, policies often select actions in regions with sparse data coverage, where the UCB may be overly conservative near well-explored safe actions, or insufficiently conservative in unexplored regions. The SSN addresses this by encoding a structural relationship: the safety status of an action depends not only on its own UCB, but on whether nearby actions are known to be safe. In other words, the SSN allows us to propagate safety across different actions.

### 4.2.2 Training

We start by considering the GP-based definition of the safe set (Sui et al., 2015) with the Euclidean distance metric for a simplified finite action space RL setting. For finite settings, the safe set at state $s$ and training step $t$ for cost function $c$, denoted as $S_{c,t}$ [†], is defined as the set of actions that are deemed to be safe at $s$:

$$S_{c,t}(s) = \bigcup_{a \in S_{c,t-1}(s)} \{a' \in \mathcal{A} \,|\, \mathrm{UCB}(s,a) + L_{Q_c}\|a - a'\|_2 \le h\}, \tag{12}$$

where $L_{Q_c}$ is the assumed Lipschitz constant for the cost Q-value $Q_c$, which can be equivalently written as

$$S_{c,t}(s) = \{a' \in \mathcal{A} \,|\, \min_{a \in S_{c,t-1}(s)} \mathrm{UCB}(s,a) + L_{Q_c}\|a - a'\|_2 \le h\}. \tag{13}$$

In order to extend this definition to the RL training setting in continuous action spaces, we make three changes. First, we define the safe set network $\hat{S}_c$ to fit to the left hand side of the inequality in (13), which allows us to redefine safety for continuous spaces. Specifically, we call a state-action pair $(s,a)$ safe if we have $\hat{S}_c(s,a) \le h$, which is defined for continuous state and action spaces. Second, analogously to how Q-functions are learned, we introduce a target network $\hat{S}_c^{\mathrm{target}}$ and use it to select safe actions in the computation of the minimum, which allows us to avoid retraining a safe set network at every time step. Third, since we cannot evaluate every action in the space for minimization, we instead check only the actions in a finite "anchor set" $\mathcal{A}_{\mathrm{anc}}$. Consequently, our training target for $\hat{S}_c(s,a')$ becomes

$$y(s,a') \leftarrow \min_{\substack{a \in \mathcal{A}_{\mathrm{anc}} \\ \hat{S}_c^{\mathrm{target}}(s,a) \le h}} \mathrm{UCB}(s,a) + L_{\hat{Q}_c}\|a - a'\|_2 \tag{14}$$

where $L_{\hat{Q}_c}$ is the Lipschitz constant for the learned cost critic.[‡] If none of the anchors are deemed safe, we use the UCB of the query point as a fallback target. We depict a simplified version of the anchor filtering, target computation and safe set network update in Figures 1b, 1c and 1d, respectively.

---

[†]In GP-based works, it is usually assumed that a "seed set" is given to initialize $S_{c,0}$. This assumption is usually needed in proofs of safety guarantees. Since we are working in a purely online setting we do not assume a seed set; however, in an offline safe RL setting this may be applicable.

[‡]It is possible for the true Q-function to be discontinuous, however since in practice we work with a neural network cost critic, we assume that we always have a continuous estimate for the true function. See the "Lipschitz assumptions" section in discussion for more details.

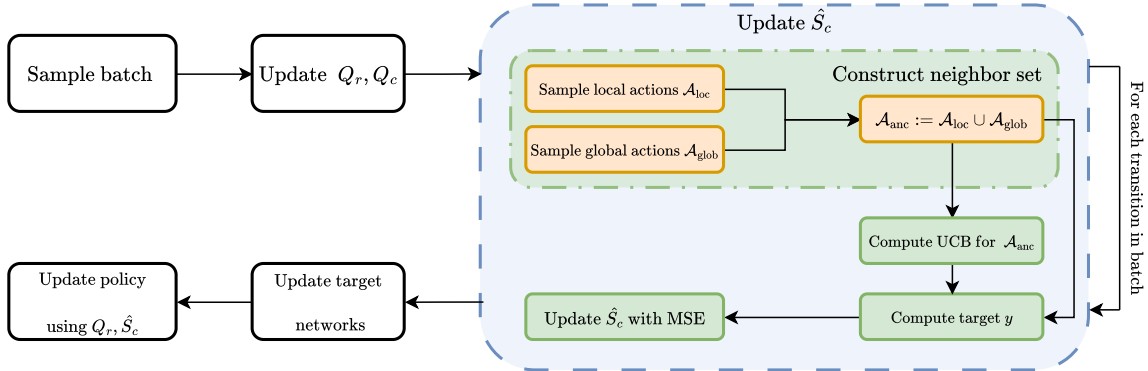

Figure 2: A high level overview of USPC.

When the safe set network $\hat{S}_c$ is fully trained to fit to this training target, we obtain a cost surrogate that is more conservative than simply using the cost critic ensemble mean $\hat{Q}_c$, as shown in Theorem 1:

**Theorem 1.** *(Conservativeness of $\hat{S}_c$) Suppose that $\hat{S}_c$ and $\hat{Q}_c$ have converged to their targets, and $\hat{Q}_c$ is L-Lipschitz. For any query state-action pair $(s, a)$, the following bound holds:*

$$\hat{Q}_c(s,a) + \min_{a' \in \mathcal{A}_{anc}^{safe}} \beta \hat{\sigma}_c(s,a') \leq \hat{S}_c(s,a) \leq \hat{Q}_c(s,a) + \min_{a' \in \mathcal{A}_{anc}^{safe}} \beta \hat{\sigma}_c(s,a') + 2L\|a - a'\|_2, \tag{15}$$

*where $\mathcal{A}_{anc}^{safe} = \{a' \in \mathcal{A}_{anc} : \hat{S}_c^{target}(s,a') \leq h\}$. Note that if the query action $a$ itself is safe, the above bound implies*

$$\hat{Q}_c(s,a) + \min_{a' \in \mathcal{A}_{anc}^{safe}} \beta \hat{\sigma}_c(s,a') \leq \hat{S}_c(s,a) \leq \mathrm{UCB}(s,a). \tag{16}$$

Note that if the standard deviation is lower bounded by some $\epsilon > 0$, Theorem 1 alongside Assumption 1 implies that $\hat{S}_c$ is also conservative compared to the true cost Q-value $Q_c$ with high probability:

**Corollary 1.** *Suppose that Assumption 1 holds, and we have $\hat{\sigma}_c(s,a) \geq \epsilon, \forall a \in \mathcal{A}$ for some $\epsilon > 0$. Then, by Theorem 1, we have*

$$Q_c(s,a) \leq \hat{S}_c(s,a) \tag{17}$$

*with probability at least $\Phi\left(\frac{\beta \epsilon \sqrt{M}}{\hat{\sigma}_c(s,a)}\right)$, where $\Phi$ denotes the standard Gaussian cumulative distribution function.*

See Appendix A.1 and A.2 for the proofs of Theorem 1 and Corollary 1, respectively.

## 4.3 Anchor Set Generation

Since the state is not directly controlled by the agent, we define propagation only in the action dimension. Thus, we train $S_c$ by varying actions for a fixed state. In order to approximate doing a pass on the entire action space, we create an anchor set with local and global samples. Local samples, denoted by $\mathcal{A}_{anc}^{loc}$ are sampled from a Gaussian input around the policy action. For deterministic policies we use a Gaussian centered around the selected action (e.g. the exploration noise distribution), while for stochastic policies the policy's action distribution output can be directly used instead. Global actions, denoted by $\mathcal{A}_{anc}^{glob}$ are sampled uniformly from the action space (assuming that it is bounded). Finally, we also include the queried point itself, since intuitively if its own UCB is below the cost threshold then the point itself can be considered safe. The anchor set for the queried state-action pair $(s, a)$ is then defined as $\mathcal{A}_{anc} := \{a\} \cup \mathcal{A}_{anc}^{loc} \cup \mathcal{A}_{anc}^{glob}$.

The anchor set allows us to efficiently approximate a full action space scan, which would only be tractable in discrete and bounded settings. In continuous domains, the global anchors $\mathcal{A}_{anc}^{glob}$ emulate the full scan of the action space by querying the regions that are not often visited by the policy. However, we also include local anchors $\mathcal{A}_{anc}^{loc}$. This is motivated by the observation that for most policy gradient methods, a small step size in

parameter updates tends to yield small changes in actions on average (although this is not guaranteed). This is especially the case for TRPO-based methods (Schulman et al., 2017), where a KL divergence constraint is explicitly used in order to limit the change in action distributions. Thus, we include the local anchors in order to get a better estimate around the existing policy behavior. Figure 1b illustrates how anchor points are sampled both locally (with respect to the policy) and globally over the action set.

The two anchor types serve complementary roles, where the global anchors achieve approximate coverage of the full action domain, whereas local anchors refine coverage near the current policy mean. We formalize these effects separately in the below results.

**Theorem 2.** *(Global anchor covering) Consider the continuous safe set $\mathcal{F} = \{a \in \mathcal{A} \,|\, \hat{S}_c^{\text{target}}(s,a) \leq h\}$ and the anchor-based safe set $\mathcal{F}_{\text{anc}} = \mathcal{F} \cap \mathcal{A}_{\text{anc}}$. Define the ideal and anchor-based SSN targets as:*

$$S_c^*(s,a) = \min_{a' \in \mathcal{F}} f(a') \qquad S_c(s,a) = \min_{a' \in \mathcal{F}_{\text{anc}}} f(a') \,, \tag{18}$$

*where $f(a') = \text{UCB}(s,a') + L\|a - a'\|_2$ is the objective and $f$ is $(L_{\text{UCB}} + L)-$Lipschitz over action space. Let $\mathcal{A}_{\text{anc}}$ consist of $K_{\text{glob}}$ points drawn uniformly from $\mathcal{A} = [a_{\min}, a_{\max}]$ with $\Delta = a_{\max} - a_{\min}$. Then, assuming $\mathcal{F}$ is connected and $\mathcal{F}_{\text{anc}} \neq \emptyset$:*

$$0 \leq S_c(s,a) - S_c^*(s,a) \leq (L_{\text{UCB}} + L)\,\epsilon_{\text{safe}}(s) \,, \tag{19}$$

*where $\epsilon_{\text{safe}}(s) = \sup_{a' \in \mathcal{F}} \inf_{a'' \in \mathcal{F}_{\text{anc}}} \|a' - a''\|_2$ is the safe covering radius, and in expectation over the random anchor draws:*

$$\mathbb{E}[S_c(s,a) - S_c^*(s,a)] = O\left(\frac{\log K_{\text{glob}}}{K_{\text{glob}}}\right). \tag{20}$$

The implication is that global anchors act as a coarse but improving proxy for scanning the entire action space, they reduce the chance that SSN misses distant safe regions because current policy does not visit them. We next show that local anchors satisfy a similar spacing bound inside a neighborhood of the policy mean.

**Proposition 1.** *(Local anchor covering near the policy mean) Under the conditions of Theorem 2, let $\mathcal{A}_{\text{anc}}$ consist of $K_{\text{loc}}$ points drawn i.i.d. from $\mathcal{N}(\mu_\pi, \sigma_\pi^2)$, clamped to $\mathcal{A}$, where $\mu_\pi$ and $\sigma_\pi^2$ denote the policy mean and variance. Let $I_r = [\mu_\pi - r, \mu_\pi + r] \subseteq \mathcal{A}$ for some $r > 0$, and assume that $\mathcal{F} \cap I_r \neq \emptyset$. Let $q_r = 2\Phi(r/\sigma_\pi) - 1$ be the probability that a draw from $\mathcal{N}(\mu_\pi, \sigma_\pi^2)$ lands in $I_r$, and $N_r \sim \text{Binom}(K_{\text{loc}}, q_r)$ be the number of local anchors falling in $I_r$. Also, redefine SSN targets over $I_r$ as:*

$$S_{c,r}^*(s,a) = \min_{a' \in \mathcal{F} \cap I_r} f(a') \qquad S_{c,r}(s,a) = \min_{a' \in \mathcal{F}_{\text{anc}} \cap I_r} f(a') \,. \tag{21}$$

*Then,*

$$0 \leq S_{c,r}(s,a) - S_{c,r}^*(s,a) \leq (L_{\text{UCB}} + L)\,\epsilon_{\text{safe}}(s) \,, \tag{22}$$

*where $\epsilon_{\text{safe}}(s) = \sup_{a' \in \mathcal{F} \cap I_r} \inf_{a'' \in \mathcal{F}_{\text{anc}} \cap I_r} \|a' - a''\|_2$ is the safe covering radius inside the ball $I_r$, and in expectation over the random anchor conditioned on $N_r = K_{\text{loc}}$:*

$$\mathbb{E}[S_{c,r}(s,a) - S_{c,r}^*(s,a) \,|\, N_r = K_{\text{loc}}] = O\left(\frac{\log K_{\text{loc}}}{K_{\text{loc}}}\right) \,. \tag{23}$$

This result suggests that local anchors provide a similar covering as in Theorem 2 inside a neighborhood of the policy mean, and therefore provide a better approximation where the policy is most likely to act, so they can improve the SSN target near the current decision region even when the global anchors are relatively sparse. We note that Proposition 1 provides a weaker guarantee than Theorem 2 because it holds only within a local neighborhood around the policy mean and is conditioned on the event that sufficiently many local anchors fall within this neighborhood. Nevertheless, it shows that local anchors improve the SSN accuracy near the policy mean.

**Remark 1.** *(Covering in higher dimensions) Theorem 2 is stated in the one-dimensional setting, where connected safe sets are intervals and the resulting bound gives an $O(\log n / n)$ rate. In higher dimensions, similar results on bounded supports suggest a dimension dependent scaling $O((\log n / n)^{1/d})$, see Section 2 of Brauchart et al. (2016) for more detail.*

---

**Algorithm 1** Safe set net update

---

1: **Input:** Batch size $B$, particle sizes $K_{\mathrm{loc}}, K_{\mathrm{glob}}$, cost critic Lipschitz constant $L$
2: **Output:** Updated safe set network parameters
3: Sample $B$ transitions $\{(s_b, a_b, r_b, c_b, s_{b+1})\}_{b=1}^{B}$ from replay buffer
4: **for** $b = 1, \ldots, B$ **do**
5:     Sample $K_{\mathrm{loc}}$ local actions $\mathcal{A}_{\mathrm{anc}}^{\mathrm{loc}} := \{a_{b,j}^{\mathrm{loc}}\}_{j=1}^{K_{\mathrm{loc}}}$ for $s_b$ as $a_{b,j}^{\mathrm{loc}} \sim \mathcal{N}(\mu_{\theta_i}, \sigma_{\theta_i}^2), \quad j = 1, \ldots, K_{\mathrm{loc}}.$
6:     Sample $K_{\mathrm{glob}}$ global actions $\mathcal{A}_{\mathrm{anc}}^{\mathrm{glob}} := \{a_{b,j}^{\mathrm{glob}}\}_{j=1}^{K_{\mathrm{glob}}}$ for $s_b$ as $a_{b,j}^{\mathrm{glob}} \sim \mathrm{Uniform}(\mathcal{A}), \quad j = 1, \ldots, K_{\mathrm{glob}}.$
7:     Set anchor set as $\mathcal{A}_{\mathrm{anc}} := \{a_b\} \cup \mathcal{A}_{\mathrm{anc}}^{\mathrm{loc}} \cup \mathcal{A}_{\mathrm{anc}}^{\mathrm{glob}}$
8:     Clamp actions in $\mathcal{A}_{\mathrm{anc}}$ to be in $[a_{\min}, a_{\max}]$
9:     Compute $\mathcal{A}_{\mathrm{anc}}^{\mathrm{safe}} := \{a' \in \mathcal{A}_{\mathrm{anc}} : \hat{S}_c^{\mathrm{target}}(s_b, a') \leq h\}$
10:     **if** $\mathcal{A}_{\mathrm{anc}}^{\mathrm{safe}} \neq \emptyset$ **then**
11:         Compute target
$$y(s_b, a_b) = \min_{a' \in \mathcal{A}_{\mathrm{anc}}^{\mathrm{safe}}} \mathrm{UCB}(s_b, a') + L\|a_b - a'\|_2$$
12:     **else**
13:         Compute target $y(s_b, a_b) = \mathrm{UCB}(s_b, a_b)$
14:     **end if**
15: **end for**
16: Update $\hat{S}_c$ with loss $\frac{1}{B} \sum_{i=1}^{B} \left( \hat{S}_c(s_i, a_i) - y(s_i, a_i) \right)^2$
17: Update $\hat{S}_c^{\mathrm{target}}$ via EMA towards $\hat{S}_c$

---

**Remark 2.** *(Mixed anchor set) The SSN target is computed by minimizing over the full anchor set $\mathcal{A}_{anc} = \{a\} \cup \mathcal{A}_{anc}^{loc} \cup \mathcal{A}_{anc}^{glob}$. When local anchors are added to a given global anchor set, or vice versa, the SSN target can only improve since (14) is performed over a larger anchor set. However, under a fixed total anchor budget, mixed anchor set is a tradeoff, where global anchors provide domain coverage whereas local anchors provide higher resolution near the policy mean. The mixed anchor set is intended to balance these two effects.*

The anchor set construction and the overall USPC algorithm can be found in Algorithm 1. USPC-adapted versions of off-policy algorithms can be found in Appendix B in detail.

## 4.4 Exploration and Conservativeness

At each iteration $t$, the ensemble of cost critics provides multiple predictions for the cost action-value function at a given state-action pair. Motivated by optimism-based methods for exploration (Sui et al., 2015; Lee et al., 2021), we construct an upper confidence bound on the cost action-value function as in (10) using Assumption 1, where $\beta$ is a hyperparameter controlling the degree of conservativeness. Intuitively, this bound increases the cost estimates in regions where ensemble disagreement is high, corresponding to areas where the learned critics are less reliable.

Unlike classical UCB methods, which use optimism to encourage exploration, we use the upper confidence bound as a pessimistic surrogate for safety evaluation. In explored regions where ensemble members agree, the confidence bound closely matches the mean estimate. In contrast, in poorly explored regions, the bound increases and discourages unsafe actions that rely on uncertain cost predictions. This ensemble-based confidence construction is simple to compute and scalable to continuous domains.

To discuss the conservativeness introduced by this construction, we first consider mild regularity assumptions on the true cost action-value function. In particular, we assume that for each fixed state, the true cost action-value function $Q_c(s, \cdot)$ is $L_{Q_c}$-Lipschitz with respect to the action:

$$|Q_c(s, a) - Q_c(s, a')| \leq L_{Q_c} \|a - a'\|_2, \qquad s \in \mathcal{S}, \ a, a' \in \mathcal{A} . \tag{24}$$

Under this regularity assumption and assuming that the true cost action-value function resides under the ensemble confidence bound, then the true cost of an action can be upper bounded by the confidence bound

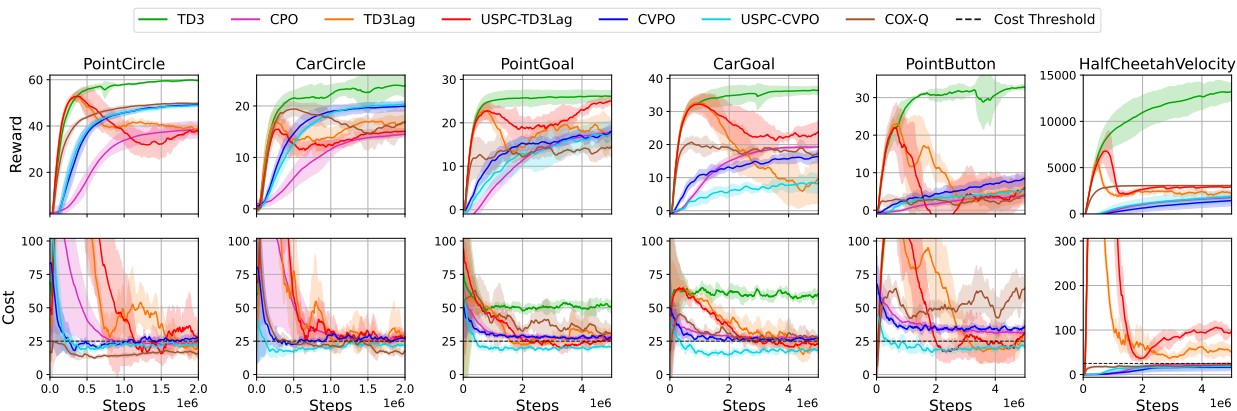

Figure 3: Training reward and cost curves for baselines and our method, each column representing a task. Solid lines represent the mean and shaded regions represent the 95% confidence interval, across 5 random seeds. Cost threshold is 25 for all tasks.

of a nearby action plus a Lipschitz correction term:

$$Q_c(s, a) \leq Q_c(s, a') + L_{Q_c} \|a - a'\|_2 \leq u_t(s, a') + L_{Q_c} \|a - a'\|_2, \qquad s \in \mathcal{S}, \ a, a' \in \mathcal{A} . \qquad (25)$$

This observation implies that the targets used to train the safe set network provide upper bounds on the true cost values whenever the conservativeness hyperparameter is picked properly. As a result, when the safe set network predicts that a state-action pair is safe, the true cost is unlikely to exceed the specified threshold, even in the presence of approximation error or limited data, which we formalize in Corollary 1.

## 5 Experiment Results

### 5.1 Setup

We use FSRL (Liu et al., 2024) and OmniSafe (Ji et al., 2024) frameworks for our implementation. OmniSafe is used for TD3, CPO, TD3Lag and USPC-TD3Lag experiments, for CVPO and USPC-CVPO we use FSRL, and finally for COX-Q (Li et al., 2026), we use the codebase provided by the authors. For our runs, we used the default applicable hyperparameters provided with these frameworks, and tuned the ones introduced by USPC (see Appendix D). For each algorithm-task pair, we report our results on 5 random seeds. Our experiments were run on a computer with an NVIDIA GeForce RTX 3090 GPU.

In our experiments, we use a subset of tasks found in Safety Gymnasium (Ji et al., 2023). Our Safety Gymnasium tasks have the following agents:

- Point: A 2-dof spherical robot with two joints; one for forward-backward motion and one for rotation. Actions contain the force in the forward-backward direction and the angular velocity for rotation.

- Car: A 3-dof robot with two parallel front wheels and one rear wheel. Actions contain the tuple of forces (in newtons) in each front wheel joint.

- HalfCheetah: An 8-dof robot consisting of 9 body parts (two legs and a body connecting them).

We test these agents on the following tasks:

- Circle: Rewards are given for following a predefined circle. Costs are incurred if the agent moves outside the boundaries (upper and lower limits on the x-axis).

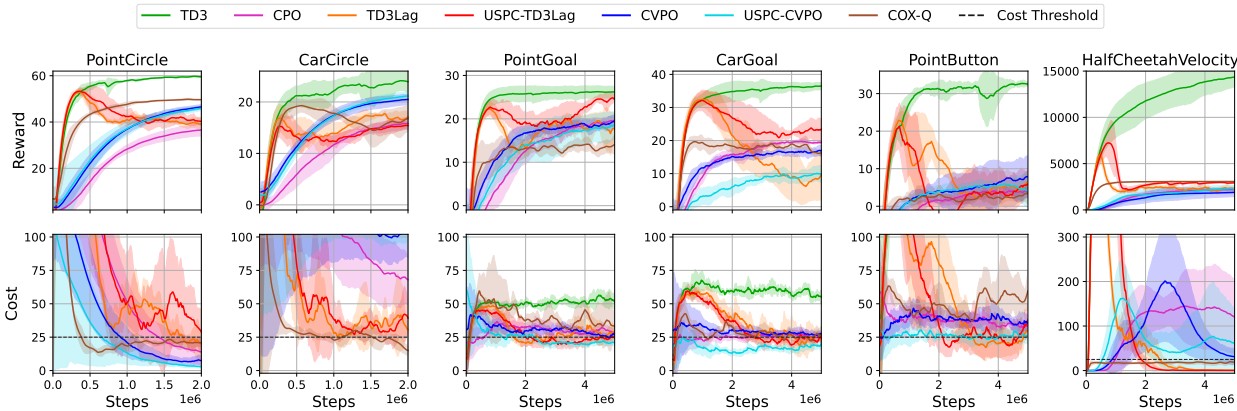

Figure 4: Evaluation reward and cost curves for baselines and our method, each column representing a task. Solid lines represent the mean and shaded regions represent the 95% confidence interval, across 5 random seeds. Cost threshold is 25 for all tasks.

- Goal: Rewards are given for moving towards a goal position. Costs are incurred for going through circular hazard zones scattered around the environment.

- Button: Rewards are given for pressing the correct goal button. Costs are incurred for going through circular hazard zones or colliding with dynamic moving obstacles in the environment.

- Velocity: Rewards are given for moving in the +x direction. Costs are incurred if the velocity for the current time step exceeds a velocity threshold.

For all tasks, we use a common episode length of 1000 and a cost threshold of 25 as our safety criterion, unless told otherwise.

## 5.2 Baselines

We run our experiments with two USPC variants alongside five model-free RL baselines. As an unsafe RL baseline, we use TD3 (Fujimoto et al., 2018), a popular off-policy actor-critic algorithm. Similarly, to be able to make a comparison with on-policy safe RL methods, we use CPO (Achiam et al., 2017) which is a trust region-based second-order algorithm. We use TD3Lag, a standard primal-dual method, and

CVPO (Liu et al., 2022b), a variational inference-based method, as our off-policy safe RL baselines where we test the effect of USPC. Finally, as a recent model-free safe RL baseline we use COX-Q (Li et al., 2026), a distributional safe RL method that also employs an ensemble-based approach.

## 5.3 Results

**Reward and cost performances.** Figures 3 and 4 show the performance of all methods in various environments in training and evaluation episodes, respectively. Table 1 shows the average reward and cost near the end of training to assess the final performance. Note that for some tasks, TD3's cost curves are not visible because it yields extremely high costs compared to safe RL algorithms (see Appendix E for full plots). We see that for most tasks, USPC variants yield lower or equal costs compared to the base algorithms, while still yielding similar rewards. Interestingly, in some tasks (PointGoal and CarGoal) USPC variants achieve similar or higher reward returns while exhibiting lower cost performance compared to their base algorithms. This behavior is consistent with our design: by using uncertainty-aware safety propagation, USPC reduces excessive conservativeness induced by noisy cost critics in well-explored regions while still encouraging safety. As a result, the policy can exploit genuinely safe, high-reward behaviors that are incorrectly penalized by standard cost critics, leading to improved reward without increasing constraint violations. We also note that

| Method | PointCircle | CarCircle | PointGoal | CarGoal | PointButton | HalfCheetahVelocity |
|---|---|---|---|---|---|---|
| TD3 | 59.8 ± 0.1
199.7 ± 1.3 | 24.0 ± 2.0
198.9 ± 8.9 | 26.1 ± 1.5
51.7 ± 4.5 | 36.4 ± 1.2
59.8 ± 3.2 | 32.8 ± 0.7
121.4 ± 2.4 | 13206.0 ± 1039.0
966.8 ± 10.4 |
| CPO | 38.4 ± 5.0
23.7 ± 2.6 | 14.4 ± 0.7
27.1 ± 3.0 | 17.8 ± 1.4
27.9 ± 1.3 | 19.3 ± 0.5
27.1 ± 1.3 | 3.7 ± 0.7
31.5 ± 2.4 | 1745.1 ± 174.9
24.4 ± 0.5 |
| TD3Lag | 39.3 ± 2.7
22.9 ± 11.9 | 16.7 ± 1.4
29.5 ± 8.0 | 17.0 ± 5.3
29.8 ± 3.8 | 9.3 ± 9.6
23.7 ± 4.9 | 5.4 ± 3.1
28.6 ± 6.4 | 2318.3 ± 413.7
54.4 ± 17.2 |
| CVPO | 49.1 ± 0.5
27.8 ± 4.7 | 19.9 ± 1.2
26.0 ± 1.0 | 17.9 ± 2.1
28.2 ± 2.2 | 16.2 ± 2.7
26.8 ± 1.4 | 8.3 ± 2.0
35.5 ± 1.4 | 1423.4 ± 812.8
16.4 ± 11.5 |
| COX-Q | **49.8 ± 0.2**
**16.3 ± 1.3** | 16.6 ± 2.8
17.2 ± 2.6 | 14.6 ± 1.8
30.2 ± 5.1 | 16.9 ± 0.8
27.0 ± 2.6 | 3.9 ± 1.9
64.1 ± 10.3 | **3067.1 ± 11.9**
**21.4 ± 5.8** |
| **USPC-TD3Lag** | 38.1 ± 1.6
29.2 ± 17.4 | 15.1 ± 0.7
28.9 ± 5.6 | 24.9 ± 1.1
26.3 ± 3.4 | **23.2 ± 3.6**
**22.9 ± 5.0** | 5.6 ± 3.8
29.1 ± 5.1 | 2888.6 ± 71.1
94.7 ± 12.8 |
| **USPC-CVPO** | 48.9 ± 0.8
23.3 ± 1.6 | **20.3 ± 0.9**
**21.4 ± 1.2** | **17.2 ± 4.4**
**20.7 ± 1.0** | 8.3 ± 4.0
18.1 ± 1.0 | **5.1 ± 0.9**
**21.1 ± 2.7** | 1865.5 ± 900.5
18.7 ± 7.4 |

Table 1: Train reward and cost (mean ± std over last 50 episodes). Unsafe task-algorithm pairs are colored purple, and safe task-algorithm pairs with best reward for each task are highlighted in **blue**. Cost threshold is 25 for all tasks.

the largest drop in reward performance occurs in HalfCheetahVelocity, which we elaborate further in the next paragraph.

**Performance on HalfCheetahVelocity.** USPC yields smaller improvements over the base algorithms on this task compared to other tasks. We attribute this to structural properties that challenge not only USPC but safe RL methods more broadly. HalfCheetah has a 6-dimensional action space compared to 2 dimensions for Point and Car, and as noted in Remark 1, anchor covering scales as $O((\log K/K)^{1/d})$, so a fixed anchor budget provides sparser coverage in higher dimensions, a challenge common to all sampling-based approaches.

Additionally, unlike the geometry-based tasks where costs arise from proximity to hazards and are largely decoupled from rewards, HalfCheetahVelocity's costs and rewards are tightly coupled: both depend directly on the agent's velocity. This leaves a narrow feasible region for all methods, and reduces the relative benefit of safety propagation. The steep dependence of cost on action also explains why a larger Lipschitz constant ($L = 10$ vs $L = 2$) is required for this task (Table 2).

**Training and evaluation cost discrepancy.** A discrepancy between training and evaluation costs can be observed across several methods. This is expected, since training and evaluation policies typically differ: during training, actions are sampled stochastically (either through learned distributions or added exploration noise), whereas during evaluation, the deterministic mean action is used. Since USPC constructs its safe set surrogate from data generated by the stochastic training policy, the safety estimates are most reliable in regions where the training policy has high probability mass. The deterministic evaluation policy may select actions outside these well-covered regions, leading to different cost outcomes. For most tasks, evaluation costs are slightly higher than training costs, which is consistent with the mean action lying near the constraint boundary while the stochastic policy averages over both safe and unsafe actions during training. The reverse occurs in HalfCheetahVelocity, where we believe USPC shifts the estimated safe region such that the mean action falls into a lower-cost part of the action space, while the stochastic training policy still samples some high-cost actions.

**Constraint violation rates.** Figure 5 shows the percentage of episodes that violate the cost constraint for all methods. Note that we omit TD3 since it violates constraints in almost all episodes on all tasks (see Appendix E). In nearly all tasks, USPC methods exhibit lower median and mean violation rates than the corresponding baselines. This reduction is particularly observed in Goal and Button tasks where USPC-CVPO achieves the lowest violation percentages among all methods by a clear margin. While COX-Q performs

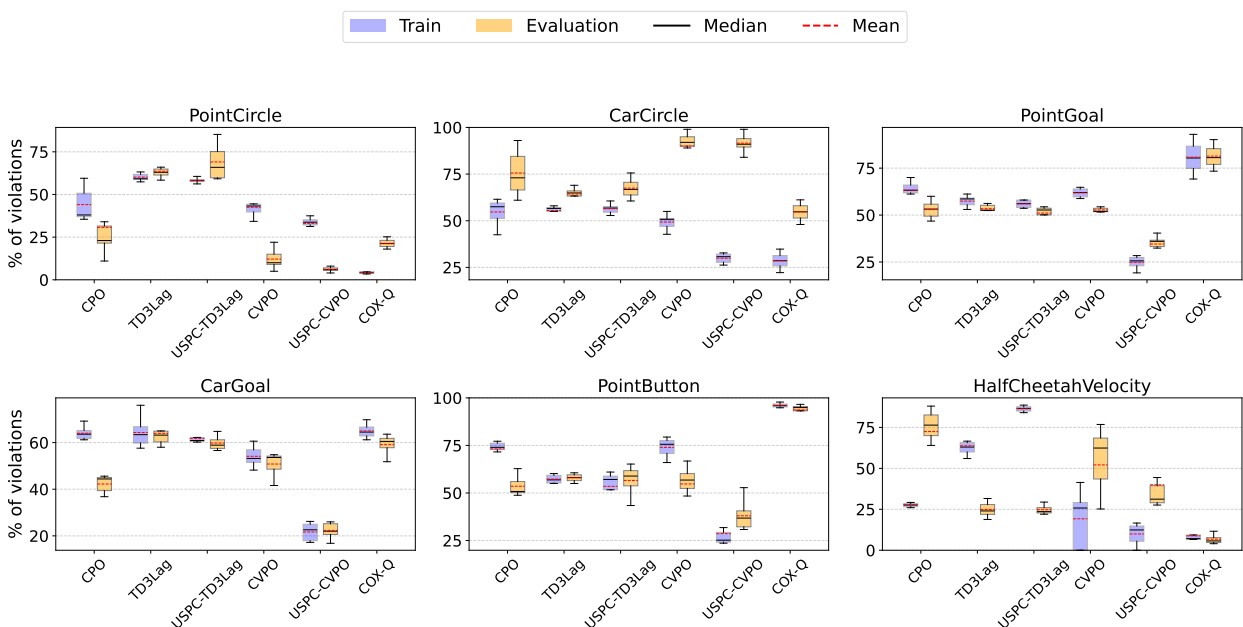

Figure 5: Box plots representing violation rates over episodes during training and evaluation episodes, for experiment runs across 5 random seeds. Shaded boxes represent an interquartile range whereas the lower and upper extremes represent the minimum and maximum, respectively.

better than the USPC-based methods on Circle and HalfCheetahVelocity tasks, it yields significantly higher violation rates on others, whereas the performance of the USPC-based methods appears more consistent across different tasks.

**Effectiveness of USPC variants.** We observe that USPC yields larger reductions in cost (and consequently lower constraint violation rates) when combined with CVPO than with TD3Lag. We believe this is because CVPO explicitly relies on accurate action-level cost estimates during its constrained E-step optimization, whereas TD3Lag only uses cost estimates through a scalar Lagrangian penalty. SSN targets are computed over an anchor set rather than a single action, which yields a quantity that is smoother and less sensitive to small changes or noise in the actions, and hence CVPO benefits more from USPC.

**Reward-cost tradeoff.** Figure 6 shows the Pareto frontier plots. We observe that in most tasks, USPC variants are at or near the Pareto frontier. USPC-CVPO in particular dominates CVPO for all tasks except HalfCheetahVelocity, where there is arguably a tie. Interestingly, for the HalfCheetahVelocity task, although USPC-TD3Lag violates the hard constraint, it still remains on the high-cost high-reward part of the frontier, which suggests that USPC may affect the reward-cost tradeoff in unexpected ways.

## 5.4 Ablation Study

We ablate our choice of Lipschitz constant $L$ and confidence bound parameter $\beta$ in the PointGoal task with USPC-CVPO, shown in Figure 7a and 7b, respectively.

**Ablating Lipschitz constant (Figure 7a).** In the $L$ ablation, we test three values of $L$: $L = 0$ which corresponds to simply using the UCB with the anchor set, $L = 2$ which is our default for the task, and $L = 5$ as a more conservative estimate. Our results show that all three values of $L$ eventually converge to similar cost returns. There is a directional trend suggesting that increasing $L$ allows the policy to become safe earlier at the cost of lower reward returns, though the wide confidence intervals make this difficult to confirm statistically. Intuitively, a lower $L$ makes it more likely for the SSN target to select an anchor point that yields a lower result than the queried point as the minimizer, which reduces the conservativeness of the policy and allows it to pursue higher rewards at the cost of higher costs, whereas a higher value of L discourages the

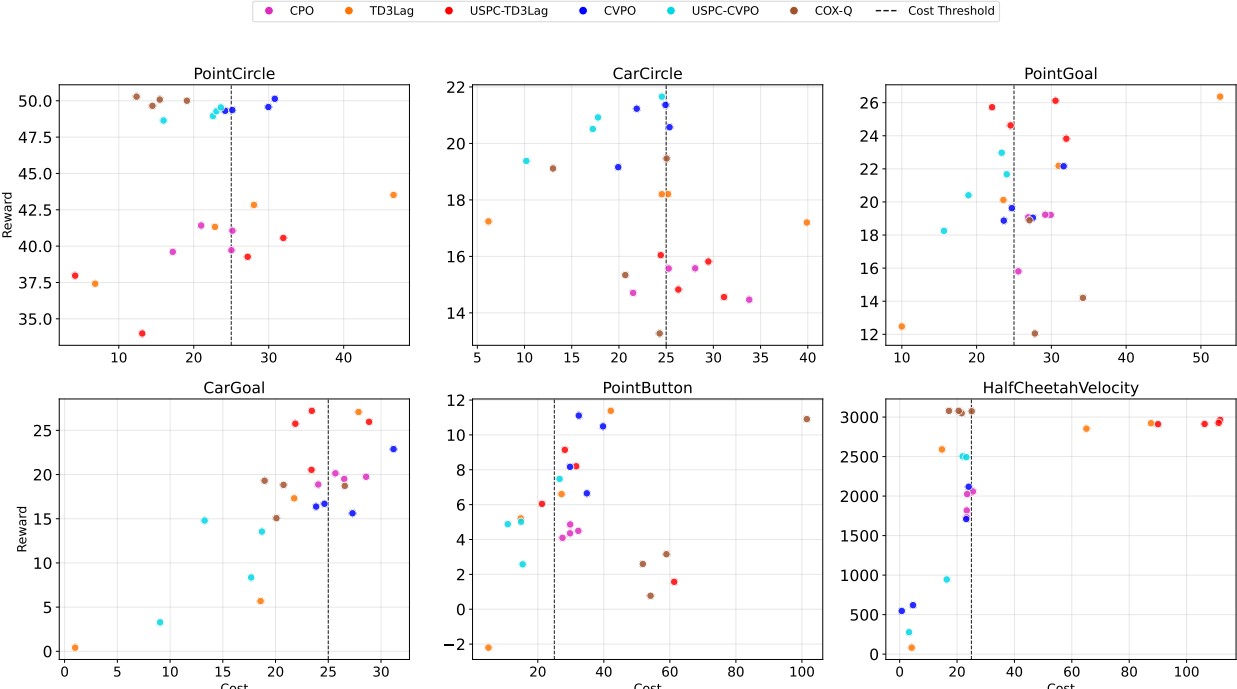

Figure 6: Pareto frontier plots of our experiment runs. Reward and cost values are averaged for the last 10 training episodes for each seed. To reduce visual clutter, best (in terms of reward-cost tradeoff) 4 seeds are taken for each method.

selection of distant anchors and limits exploration within a smaller feasible region. Notably, $L = 0$ still yields strong performance, which suggests that the primary benefit of USPC comes from the safety propagation mechanism itself (learning an SSN that aggregates cost estimates across multiple anchors) rather than from the Lipschitz distance term specifically. The distance term acts as a regularizer that discourages the SSN target from selecting anchors far from the query point, which can accelerate convergence to safe behavior but is not strictly necessary for the method to be effective.

In the limiting case of $L \to \infty$, the distance term dominates for all anchors except the query point itself, so the SSN target collapses to the UCB at the query point. This corresponds to the UCB-CVPO variant in Figure 10, which achieves similar cost but substantially lower reward. Moreover, UCB-CVPO's cost starts near the threshold and remains low throughout training, suggesting that it is overly conservative from the outset and suppresses exploration. In contrast, USPC-CVPO begins with higher costs and follows a gradual decline, indicating that the propagation mechanism allows the agent to explore more freely early in training while still converging to safe behavior. This confirms that intermediate values of $L$ allow USPC to achieve a better reward-cost tradeoff by leveraging safety information from nearby actions rather than applying uniform pessimism.

**Ablating confidence interval width (Figure 7b).** In the $\beta$ ablation, we again test three values of $\beta$: $\beta = 0$ which corresponds to using the ensemble mean instead of UCB, $\beta = 2$ which is our default, and $\beta = 5$ for a more conservative UCB. We see that there is a clear trend: A higher value of $\beta$ results in cost returns converging at lower values, at the cost of lower rewards. For $\beta = 0$ we see that although the converged reward is higher, cost converges around the threshold, which means that the constraint is violated (albeit slightly) in most episodes. On the other hand, $\beta = 5$ yields a very safe policy but the rewards are also much lower. $\beta = 2$ provides a good balance of rewards and safety.

**Ablating anchor set size and distribution (Figure 8).** We ablate anchor set size and distribution to test the effect of anchor set parameters. For the PointGoal task, we observe that the size of the anchor set does not seem to significantly affect the performance, likely since $\geq 16$ anchors are enough to cover the action space

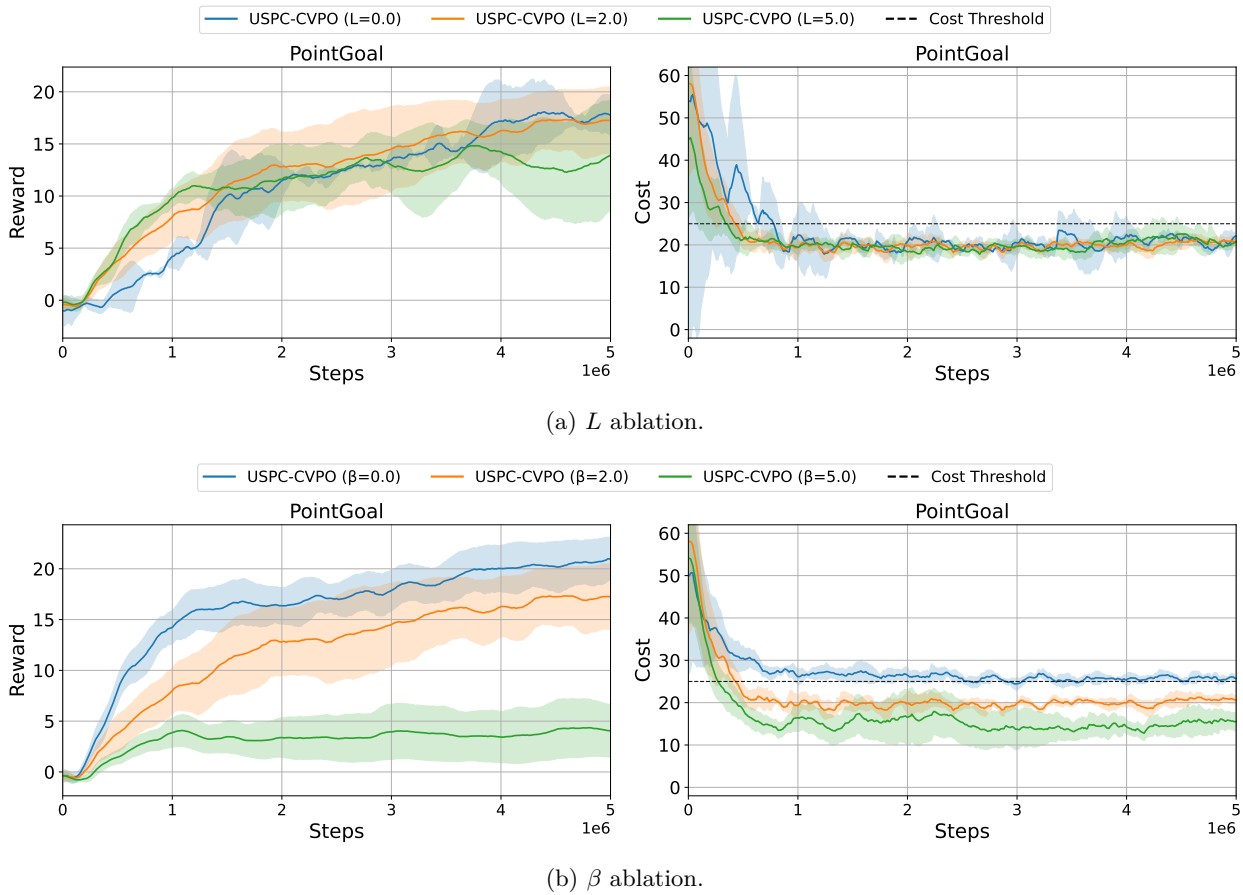

(a) *L* ablation.

(b) *β* ablation.

Figure 7: Ablation studies for USPC-CVPO on PointGoal (training episodes) for Lipschitz constant and confidence interval width. (a) Effect of the Lipschitz constant *L*. (b) Effect of the uncertainty scaling parameter *β*.

sufficiently for the Point agent (see Theorem 2 and Proposition 1). However, given enough computational resources, it might be beneficial to still use higher anchor set sizes for slightly faster convergence, since it should allow smoother targets to train the SSN with. As for the anchor set distribution we ablate the usage of global and local anchors. The results show that using global anchors only yields similar results as using both types of anchors together. On the other hand, using local anchors alone appears to have slightly lower average reward and cost. This may be because local (Gaussian) distribution is less likely to hit potential minimizers far away from the policy mean, so the computed SSN targets are slightly higher. However, since the local distribution also has support over the entire action space, these potential minimizers should eventually be discovered as different anchor sets are sampled during training, which reduces the difference in performance.

**Ablating ensemble size (Figure 9).** We ablate over our ensemble size $M$. Note that the $M = 1$ case corresponds to simply using a regular cost critic instead of the UCB in the computation of SSN targets. We observe that for $M = 1$, although the rewards are similar to our default choice of $M = 6$, the cost is much higher, which suggests that uncertainty estimation is crucial for USPC. For higher $M$, we observe that they all converge to a similar cost level, hence we argue that higher ensemble sizes have diminishing returns on the "accuracy" of the standard deviation.

**Ablating safety propagation (Figure 10).** Finally, we ablate our safety propagation mechanism in its entirety by removing the anchor set. UCB-CVPO corresponds to the CVPO algorithm, with the only difference being that the cost critic is replaced with the UCB from a cost critic ensemble (trained the same

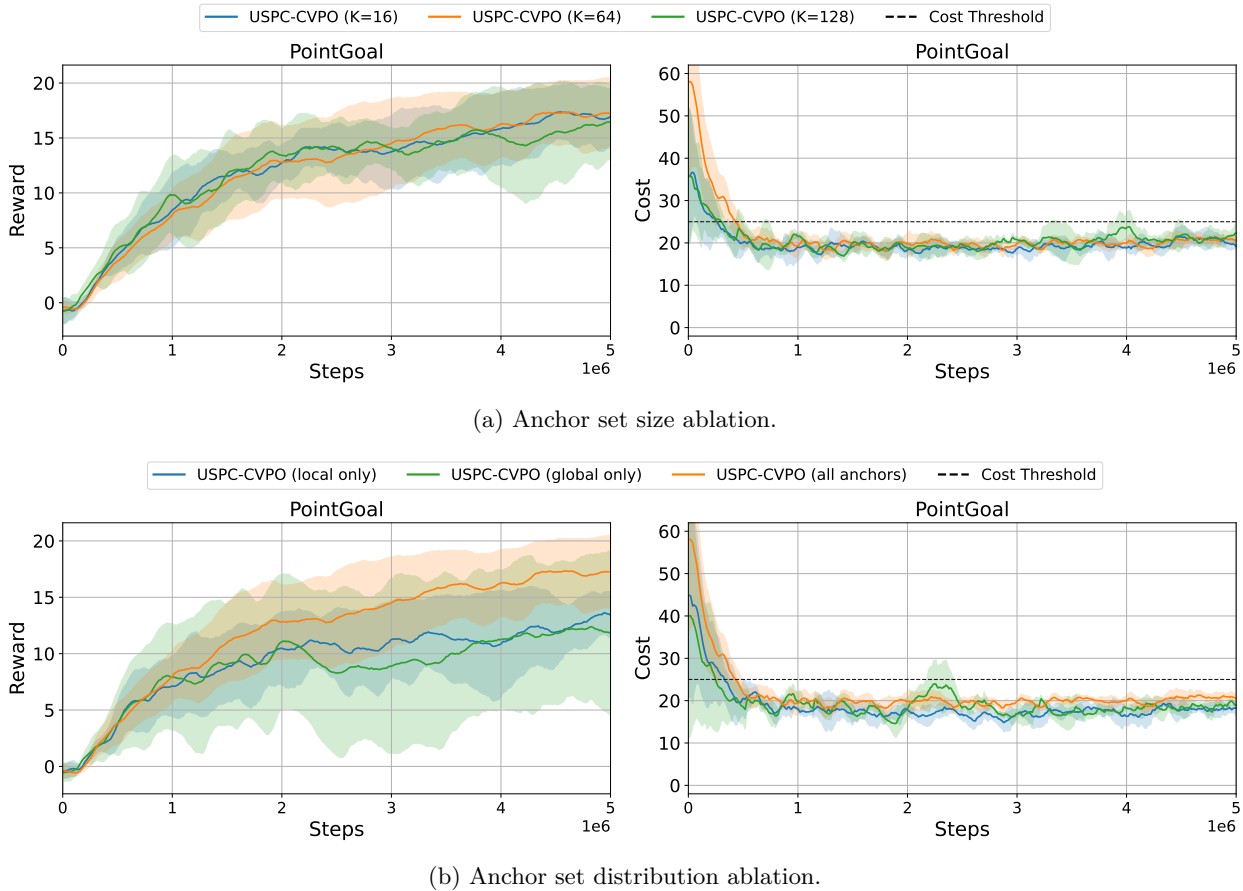

(a) Anchor set size ablation.

(b) Anchor set distribution ablation.

Figure 8: Ablation studies for USPC-CVPO on PointGoal (training episodes) for anchor set parameters. (a) Effect of the anchor size. Each run has equal amount of local and global anchors. (b) Effect of anchor set distribution. Each variant contains the same amount of total anchor points (64).

way as in USPC). We observe that while both achieve a similar cost level, USPC-CVPO has significantly higher rewards. We argue that this is because using UCB alone introduces blind conservativeness which can severely hinder exploration. On the other hand, the USPC mechanism applies selective conservativeness for states that are highly underexplored or certainly unsafe, while reducing the impact of uncertainty by propagating safety information from other actions in the same state if those actions are known to be safe.

## 6 Discussion

In this section, we elaborate on our design choices and aspects of our proposed method. We also consider our method's limitations and discuss future work.

**Approximating the uncertainty with neural networks.** A common limitation of previous GP-based works is that for higher dimensional inputs, GPs tend to scale poorly in terms of computational cost for high dimensions and cannot be easily applied to continuous spaces, which is why we propose a neural network-based uncertainty approximation, which does not suffer from these issues. Common choices include Bayesian neural networks (Neal, 2012; Blundell et al., 2015), Monte Carlo dropout (Gal & Ghahramani, 2016), and ensemble methods (Osband et al., 2016; Lakshminarayanan et al., 2017). We chose to use an ensemble approach to estimate uncertainty in the cost returns as it usually requires the least amount of modifications in the original method, since many reinforcement learning methods already use clipped double Q-learning (Fujimoto et al., 2018) while training critics which already introduces multiple critics

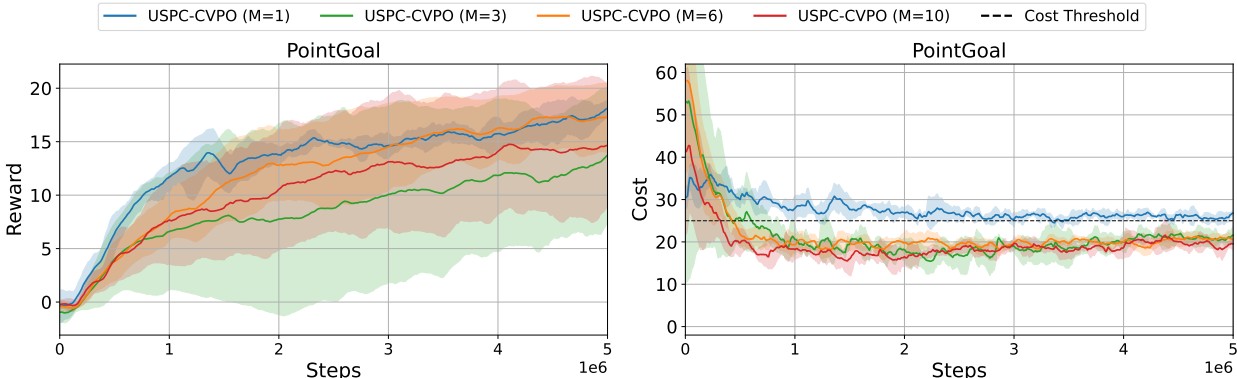

Figure 9: Ablation study for USPC-CVPO on PointGoal (training episodes) for the ensemble size.

estimating the cost return and could easily be extended to an arbitrary number of critic networks. However, it should be noted that taking the minimum over target networks to obtain a common training target breaks the independence assumption (Assumption 1), so in our implementation we compute separate targets for each ensemble member. Additionally, since ensemble diversity arises solely from random initialization while all members share identical training data and batch ordering, uncertainty may be underestimated in regions where all members converge to similarly biased predictions due to shared architecture or systematic data coverage gaps. A practical mitigation is to train ensemble members on different data subsets via bootstrapping (Lakshminarayanan et al., 2017), which increases diversity at the cost of weakening the i.i.d. assumption underlying our CLT-based confidence interval construction (Assumption 1). Investigating this tradeoff between theoretical grounding and practical calibration is an interesting direction for future work.

Finally, we note that USPC targets epistemic uncertainty in the cost critic specifically; it does not address uncertainty in transition dynamics or reward functions, which in our model-free setting are handled implicitly through sampling rather than learned models. Extending the framework to explicitly account for these additional sources of uncertainty, for example by combining USPC with model-based uncertainty quantification, is another promising direction.

**Computing SSN targets.** In the computation of SSN targets, an alternative approach is to perform gradient-based inner loop optimization to approximately solve the optimization problem in Equation 13 over the entire action space instead of an anchor set. In theory, this approach would yield more stable and accurate targets. However, in our early experiments we observed that this approach significantly increased the training time while not providing any noticeable benefit in terms of reward or cost returns. Thus, we use our anchor set-based approach that is computationally much cheaper.

We include the queried point itself in the anchor set for our SSN calculation, which has been proposed by Sui et al. (2015). The objective at the queried point collapses to the UCB at that point (since the distance term becomes 0), which means that we may consider the queried point without "certifying" it using another known safe point. We use this approach mainly because it reduces the reliance on the assumed Lipschitz constant, which in our work is a hyperparameter and may not always be chosen accurately.

**Using a safe set network to estimate the safe set.** USPC uses a neural network-based critic, the SSN, to estimate the safe set. This adds another component that needs to be learned to the training process, and in theory one could compute the SSN targets and use the targets directly in place of the cost critics. However, we use the SSN for two reasons, analogous to the use of critics in RL in general. First, the targets themselves can be very high-variance and destabilize training, since the value of the minimum depends greatly on the sampled anchor points. On the other hand, a neural network like SSN can learn a much lower variance and more generalizable estimate over time. Second, we would like to be able to extract useful gradients from our safe set estimator, which is necessary for many RL algorithms (for example, the policy gradient itself is the gradient of critics). Although the targets are also differentiable by Danskin's Theorem (Bertsekas, 1997),

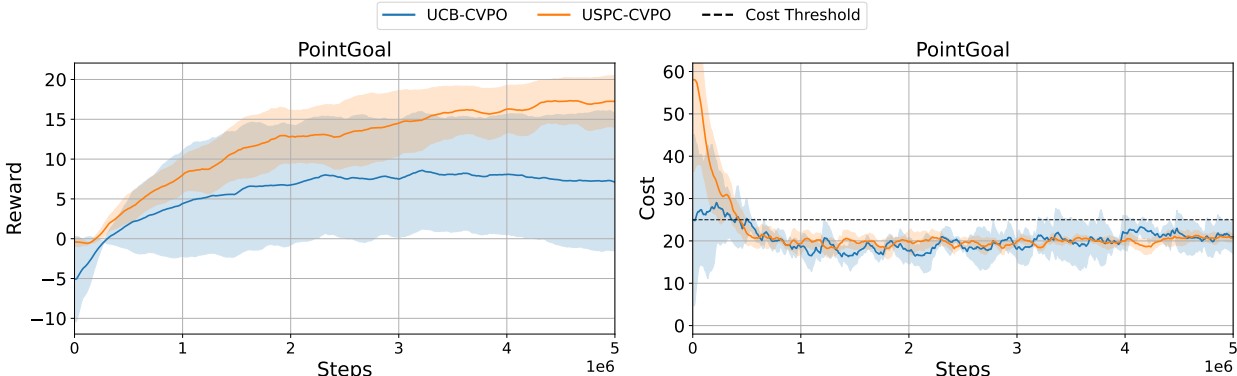

Figure 10: Ablation study for USPC-CVPO on PointGoal (training episodes) for the safety propagation mechanism.

the gradients are stochastic and similarly high-variance, and as such unsuitable for stable policy training. The SSN provides a clean differentiable surrogate through which policy gradients can flow reliably.

**Relationship to safe Bayesian optimization.** Our method is related to safe Bayesian optimization, especially methods such as Sui et al. (2015); Berkenkamp et al. (2023), which combine uncertainty-aware confidence bounds with safety propagation under a Lipschitz assumption. This work can be viewed as a scalable version of this framework in reinforcement learning. The ensemble of cost critics plays the role of an uncertainty estimator and the SSN approximates the safety propagation in continuous state-action spaces. Unlike safe Bayesian optimization, our method must account for sequential decision-making, function approximation, and nonstationary data distributions caused by policy learning. By embedding the safe set construction directly into an actor-critic framework, we preserve the core safety principles of safe Bayesian optimization while extending them to high-dimensional reinforcement learning problems with continuous state-action spaces.

**Lipschitz assumptions.** Our work assumes that the *learned* cost critic $\hat{Q}_c$ is globally $L$-Lipschitz with respect to actions for any fixed state, for some $L$. This assumption does not hold for the *true* $Q_c$ in general, since the cost function $c$ in our tasks can be discontinuous. However, it does hold for $\hat{Q}_c$ since in practice we use neural networks that have finite linear layers and weights (and therefore finite operator norms) and Lipschitz activation functions (most activation functions such as ReLU and hyperbolic tangent are Lipschitz), thus $\hat{Q}_c$ itself is also Lipschitz. However, it is difficult to get a value for $L$. One option is to derive a value mathematically. Although some previous works get an estimate for $L$ using the neural network's architecture (Virmaux & Scaman, 2018; Bjorck et al., 2021), we do not wish to make any further assumptions about the critic networks, so we consider these out of scope of our work. Another option is to estimate the value in a data-driven fashion, such as by maximizing a finite-difference slope estimate or the gradient itself over the anchor set. However, a data-driven approach often increases computational costs further, and more importantly destabilizes the training by moving the targets for SSN. Therefore, we use the third approach, which is treating $L$ as a hyperparameter. Since obtaining a tight value is already difficult, having $L$ be a hyperparameter allows us to tune it for the desired balance of safety versus reward performance, and results in a more stable SSN.

**Effect of substituting SSN on safety constraints.** Substituting SSN in place of the cost critic improves safety by modifying the constraint in the underlying optimization problem. Specifically, as discussed in Section 4.4, SSN does not provide a strict upper-bound for the cost critic; it upper-bounds the cost critic with high probability instead. What this means is that substituting SSN converts the optimization problem from

$$\pi^* \in \arg\max_{\pi} \ \mathbb{E}_{s_0 \sim \rho_0, \ a_0 \sim \pi(\cdot|s_0)}\big[Q_r(s_0, a_0)\big] \tag{26}$$

$$\text{s.t.} \ \mathbb{E}_{s_0 \sim \rho_0, \ a_0 \sim \pi(\cdot|s_0)}\big[Q_c(s_0, a_0)\big] \le h$$

to

$$\pi^* \in \arg\max_{\pi} \; \mathbb{E}_{s_0 \sim \rho_0, \, a_0 \sim \pi(\cdot|s_0)}\big[Q_r(s_0, a_0)\big] \tag{27}$$
$$\text{s.t.} \;\; \mathbb{P}\big(\mathbb{E}_{s_0 \sim \rho_0, \, a_0 \sim \pi(\cdot|s_0)}\big[Q_c(s_0, a_0)\big] \le h\big) \ge 1 - \delta$$

for some $\delta$. In other words, the substitution of SSN changes the original expected cost constraint into a high-probability constraint on the expected cost. Intuitively, and as established in Corollary 1, for sufficiently high $\beta$, the feasible region of (27) will be a subset of the feasible region of (26) with high probability, meaning that a policy that is feasible for (26) is not necessarily feasible for (27). However, empirical results show that the USPC variants mostly have lower cost returns and constraint violation rates (see Figures 3 and 4), so we argue this does not pose a problem in practice.

**Conservativeness-performance trade-off.** Naturally, increasing conservativeness (i.e., shrinking the feasible region) by substituting our SSN for the cost critic (see Theorem 1) results in worse reward returns. This is also reflected in the experiment results (Figures 3-4): USPC variants of algorithms tend to have lower reward returns compared to non-USPC ones. However, the added conservativeness also results in lower constraint violation rates (both in training and evaluation episodes), fewer training episodes until the policy becomes safe, and overall lower cost returns (meaning that violation magnitudes are also smaller even for unsafe episodes). Since our main goal is to handle hard constraints even during training, we believe that this is an acceptable trade-off, although for some tasks the degree of conservativeness can be tuned (via the parameter $\beta$) to obtain higher task rewards if a better reward return is desired.

**Limitations and future work.** Like all methods, our method introduces limitations (other than what is inherited from the base safe RL algorithm used), some of which we discuss here. First is the introduction of additional models that need to be learned: an additional SSN needs to be trained, and we also assume the cost critics are neural network ensembles rather than single or double Q-networks. This naturally increases computational costs; training with our method takes around twice as long in our setup compared to the base algorithms. Training the SSN with targets computed using the learned critics also introduces some instability. Alternative model architectures (e.g., Bayesian neural networks or MC-dropout as previously discussed) and approaches to obtaining SSN targets can help alleviate these issues.

Additionally, as discussed in Section 5.3, USPC's effectiveness depends on the anchor set providing sufficient coverage of the action space. Since covering quality degrades with dimensionality (Remark 1), the method is best suited for low-to-moderate dimensional action spaces with a fixed anchor budget. The method also yields smaller improvements when the cost and reward signals are tightly coupled, as in velocity-constrained tasks, since the feasible region of safe yet rewarding actions becomes narrow and leaves less room for safety propagation to provide benefit. Adaptive anchor allocation strategies or structured sampling that accounts for the reward-cost geometry could help address both of these limitations.

Another limitation of our method is the fixed $\beta$ in our confidence intervals. In safe BO, it is often assumed that $\beta$ is a function of step $t$ rather than a constant value, which is used to derive useful theoretical results such as a regret bound. In our case, a varying $\beta$ can lead to instability issues due to moving targets for SSN, thus we treat $\beta$ as a constant hyperparameter. However, we believe that a tunable $\beta$ may have interesting theoretical and practical implications for deep safe RL as well, which we leave for future work.

Finally, in this work, we train our SSN for cost Q-value critics. However, a sizable portion of safe RL algorithms contain cost state-value critics ($\hat{V}_c$) instead. It is also possible to train an SSN for $\hat{V}_c$, by defining

$$\hat{S}_c^{\text{state}}(s') \leftarrow \min_{\substack{s \in \mathcal{S}_{\text{anc}} \\ \hat{S}_c^{\text{target}}(s) \le h}} \text{UCB}(s) + L_{\hat{V}_c}\|s - s'\|_2^2. \tag{28}$$

However, $\mathcal{S}_{\text{anc}}$ cannot be obtained in the same way as $\mathcal{A}_{\text{anc}}$. While we assume that the agent can take any action in its action space, it cannot directly select the states it will be at in the next time step. As such, $\mathcal{S}_{\text{anc}}$ must be constructed using a dynamics model to make sure that only reachable states are selected. While this approach is more in the model-based territory which is out of our scope, it can be an interesting research direction.

## 7    Conclusion

In this work, we introduced Uncertainty-Aware Safety Propagation Critics (USPC), a model-free approach for safe reinforcement learning that constructs conservative cost surrogates by combining ensemble-based epistemic uncertainty estimation with Lipschitz-based safety propagation. By training a scalable safe set network that approximates a pessimistic cost action–value function, USPC enables safety information to generalize beyond sampled actions in continuous state–action spaces. Our proposed safe set approximation may also be used to extend some existing safe Bayesian optimization methods to continuous domains. Our method can be seamlessly integrated into existing off-policy safe actor–critic algorithms without altering their core optimization procedures. Empirical results on multiple Safety Gymnasium benchmarks demonstrate that, for most tasks, USPC reduces both the frequency and magnitude of constraint violations during training and evaluation while maintaining competitive reward performance. These results suggest that uncertainty-aware safety propagation is a practical and effective mechanism for improving robustness in model-free safe reinforcement learning under limited or unreliable cost estimates.

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

# A  Proofs of Theoretical Results

## A.1  Proof of Theorem 1

*Proof.* For the lower bound, using the fact that $\hat{Q}_c$ is $L$-Lipschitz, we have

$$
\begin{aligned}
\hat{S}_c(s,a) &= \min_{a' \in \mathcal{A}_{\text{anc}}^{\text{safe}}} \hat{Q}_c(s,a') + \beta \hat{\sigma}_c(s,a') + L\|a - a'\|_2 \\
&\geq \min_{a' \in \mathcal{A}_{\text{anc}}^{\text{safe}}} \hat{Q}_c(s,a) + \beta \hat{\sigma}_c(s,a') \\
&= \hat{Q}_c(s,a) + \min_{a' \in \mathcal{A}_{\text{anc}}^{\text{safe}}} \beta \hat{\sigma}_c(s,a').
\end{aligned}
$$

For the upper bound, we similarly obtain

$$
\begin{aligned}
\hat{S}_c(s,a) &= \min_{a' \in \mathcal{A}_{\text{anc}}^{\text{safe}}} \hat{Q}_c(s,a') + \beta \hat{\sigma}_c(s,a') + L\|a - a'\|_2 \\
&\leq \min_{a' \in \mathcal{A}_{\text{anc}}^{\text{safe}}} \hat{Q}_c(s,a) + \beta \hat{\sigma}_c(s,a') + 2L\|a - a'\|_2 \\
&= \hat{Q}_c(s,a) + \min_{a' \in \mathcal{A}_{\text{anc}}^{\text{safe}}} \beta \hat{\sigma}_c(s,a') + 2L\|a - a'\|_2.
\end{aligned}
$$

$\square$

## A.2  Proof of Corollary 1

*Proof.* Using the lower bound on the standard deviation on Theorem 1, we have

$$
\begin{aligned}
\hat{Q}_c(s,a) + \beta \epsilon &\leq \hat{Q}_c(s,a) + \min_{a' \in \mathcal{A}_{\text{anc}}^{\text{safe}}} \beta \hat{\sigma}_c(s,a') \\
&\leq \hat{S}_c(s,a)
\end{aligned}
$$

By Assumption 1,

$$
\hat{Q}_c(s,a) + \beta \epsilon \sim \mathcal{N}\left( Q_c(s,a) + \beta \epsilon, \frac{\hat{\sigma}_c^2(s,a)}{M} \right)
$$

$$
\implies \mathbb{P}(\hat{Q}_c(s,a) + \beta \epsilon \geq Q_c(s,a)) = \Phi\left( \frac{\beta \epsilon \sqrt{M}}{\hat{\sigma}_c(s,a)} \right)
$$

$$
\mathbb{P}(\hat{S}_c(s,a) \geq Q_c(s,a)) \geq \Phi\left( \frac{\beta \epsilon \sqrt{M}}{\hat{\sigma}_c(s,a)} \right)
$$

$\square$

### A.3 Proof of Lemma 1

*Proof.* Since each $\hat{f}_i$ is $L$-Lipschitz, their mean $\hat{Q}_c$ is also $L$-Lipschitz. Now define $g_i(s,a) := \hat{f}_i(s,a) - \hat{Q}_c(s,a)$ (thus $g_i$ is $2L$-Lipschitz). For the (unbiased) sample variance $\hat{\sigma}_c^2(s,a) = \frac{1}{M-1}\sum_j g_j^2(s,a)$, we have

$$
\begin{aligned}
\left|\hat{\sigma}_c^2(s,a) - \hat{\sigma}_c^2(s,a')\right| &= \frac{1}{M-1}\left|\sum_j \left(g_j^2(s,a) - g_j^2(s,a')\right)\right| \\
&= \frac{1}{M-1}\left|\sum_j \left(g_j(s,a) - g_j(s,a')\right)\left(g_j(s,a) + g_j(s,a')\right)\right| \\
&\leq \frac{1}{M-1}\sum_j |g_j(s,a) - g_j(s,a')|\,|g_j(s,a) + g_j(s,a')| \\
&\leq \frac{1}{M-1}\sum_j 2L\|a-a'\|_2\,|g_j(s,a) + g_j(s,a')| \\
&\leq \frac{1}{M-1}\sum_j 4L\|a-a'\|_2 \sup_u \max_i |g_i(s,u)| \\
&= \frac{M}{M-1}4LD\|a-a'\|_2.
\end{aligned}
$$

Now consider $h(t) = \sqrt{t}$. By the mean value theorem, for $x,y \geq 0$, we have $\left|\sqrt{x} - \sqrt{y}\right| \leq \frac{1}{2\sqrt{\xi}}|x-y|$ for some $\xi \in (\min(x,y), \max(x,y))$. Since $\xi \geq \min(x,y)$,

$$
\frac{1}{\sqrt{\xi}} \leq \frac{1}{\sqrt{\min(x,y)}} \implies \left|\sqrt{x} - \sqrt{y}\right| \leq \frac{|x-y|}{2\sqrt{\min(x,y)}}.
$$

Inserting $x = \hat{\sigma}_c^2(s,a) + \epsilon^2$ and $y = \hat{\sigma}_c^2(s,a') + \epsilon^2$, we have $\min(x,y) \geq \epsilon^2$, thus

$$
\begin{aligned}
|\hat{\sigma}_c(s,a) - \hat{\sigma}_c(s,a')| &\leq \frac{1}{2\epsilon}\left|\hat{\sigma}_c^2(s,a) - \hat{\sigma}_c^2(s,a')\right| \\
&\leq \frac{2LD}{\epsilon}\frac{M}{M-1}\|a-a'\|_2
\end{aligned}
$$

Finally, using the Lipschitzness of the ensemble mean and standard deviation and the triangle inequality, we obtain

$$
\begin{aligned}
|\text{UCB}(s,a) - \text{UCB}(s,a')| &= \left|\hat{Q}_c(s,a) + \beta\hat{\sigma}_c(s,a) - \hat{Q}_c(s,a') - \beta\hat{\sigma}_c(s,a')\right| \\
&\leq \left|\hat{Q}_c(s,a) - \hat{Q}_c(s,a')\right| + |\beta\hat{\sigma}_c(s,a) - \beta\hat{\sigma}_c(s,a')| \\
&\leq L\|a-a'\|_2 + \beta\left(\frac{2LD}{\epsilon}\frac{M}{M-1}\right)\|a-a'\|_2 \\
&= L\left(1 + \frac{2\beta D}{\epsilon}\frac{M}{M-1}\right)\|a-a'\|_2.
\end{aligned}
$$

$\square$

### A.4 Proof of Theorem 2

*Proof.* Since $\mathcal{F}_{\text{anc}} \subseteq \mathcal{F}$, minimizing over a smaller set can only increase the value, hence:

$$
S_c(s,a) \geq S_c^*(s,a) .
$$

For the upper bound, let $a^* \in \arg\min_{a' \in \mathcal{F}} f(a')$, so $f(a^*) = S_c^*(s,a)$. Since $\mathcal{F}_{\text{anc}} \neq \emptyset$, let $\tilde{a} = \arg\min_{a'' \in \mathcal{F}_{\text{anc}}} \left\|a^* - a''\right\|_2$. By definition of $\epsilon_{\text{safe}}$, we have $\left\|\tilde{a} - a^*\right\|_2 \leq \epsilon_{\text{safe}}(s)$. Since $\tilde{a} \in \mathcal{F}_{\text{anc}}$,

$$
S_c(s,a) \leq f(\tilde{a}) \leq f(a^*) + (L_{\text{UCB}} + L)\left\|a^* - \tilde{a}_2\right\| \leq S_c^*(s,a) + (L_{\text{UCB}} + L)\,\epsilon_{\text{safe}}(s),
$$

where the second inequality uses $(L_{\text{UCB}} + L)-$Lipschitzness of $f$.

We now bound the expected covering radius using the distribution of the maximum spacing. Consider $n$ points $X_1, \dots, X_n$, where $X_i \sim \text{Uni}(a_{\min}, a_{\max})$ with range $\Delta = a_{\max} - a_{\min}$. These points create $n+1$ spacings $D_0, \dots, D_n$ including boundary gaps, where $D_i = X_{i+1} - X_i$, with $D_0 = X_1 - a_{\min}$ and $D_n = a_{\max} - X_n$. These gaps satisfy $\sum_i D_i = \Delta$ and $\epsilon_{\text{safe}}(s) \leq \max_i D_i$. The spacings are exchangeable and uniform on the simplex with density $n!$ when scaled back to $[0, 1]$. By Darling (1953) and Pyke (1965), we can write the CDF of $M_n = \max_i D_i$ as:

$$\mathbb{P}(M_n \leq t) = \sum_{k=0}^{\lfloor 1/t \rfloor} (-1)^k \binom{n+1}{k} (1 - kt)^n$$

and the expectation as:

$$\mathbb{E}[M_n] = \frac{1}{n+1} \sum_{k=1}^{n+1} \frac{1}{k} \ .$$

Scaling back to $\mathcal{A}$ and substituting into the equation we have before, we get:

$$\mathbb{E}[S_c(s, a) - S_c^*(s, a)] \leq \Delta \cdot \frac{L_{\text{UCB}} + L}{K_{\text{glob}} + 1} \sum_{k=1}^{n+1} \frac{1}{k} = O\left(\frac{\log n}{n}\right) \ .$$

$\square$

## A.5 Proof of Proposition 1

*Proof.* For clarity, we treat the case where $I_r \subseteq \mathcal{A}$ so that no boundary clipping occurs. If the neighborhood is clipped by the boundary, i.e., $I_r = [\mu_\pi - r, \mu_\pi + r] \cap \mathcal{A}$, $q_r$ then can be written as $\mathbb{P}(X \in I_r)$, the rest of the results only change in constants.

The deterministic bound follows by the same argument as in Theorem 2, applied to $\mathcal{F} \cap I_r$ instead of $\mathcal{F}$ and $\mathcal{F}_{\text{anc}} \cap I_r$ instead of $\mathcal{F}_{\text{anc}}$. Conditional on $N_r = n$, the $n$ local anchors falling in $I_r$ are i.i.d. from the conditional density $\phi(a)/q_r$, where $\phi(\cdot)$ is the Gaussian PDF. Unlike the global case, these points are not uniformly distributed on $I_r$. We reduce to the uniform case via a CDF transform.

Let

$$G(a) = \mathbb{P}(X \leq a \,|\, X \in I_r) = \frac{\Phi\left(\frac{a - \mu_\pi}{\sigma_\pi}\right) - \Phi\left(\frac{-r}{\sigma_\pi}\right)}{q_r} \ ,$$

which allows us to map $I_r$ onto $[0, 1]$.

Let $X_1, \dots, X_n$ be the local anchor points that fell in $I_r$. Then, $\{Y_i\}_{i=1}^n$ are $n$ i.i.d. $\text{Uni}[0, 1]$ with $Y_i = G(X_i)$. $\{Y_i\}_{i=1}^n$ form spacings $D_i^Y = Y_{i+1} - Y_i$, with $D_0^Y = Y_1$ and $D_n^Y = 1 - Y_n$. Similarly, we have $D_i^X = X_{i+1} - X_i$, with $D_0^X = X_1 - (\mu_\pi - r)$ and $D_n^X = (\mu_\pi + r) - X_n$. This gives us $\epsilon_{\text{safe}}(s) \leq \max_i D_i^X$.

Now, we have to relate $\max_i D_i^X$ to $\max_i D_i^Y$. Since $G$ is monotonically increasing on $I_r$, $G^{-1}$ exists and is differentiable on $(0, 1)$ with $(G^{-1})'(v) = 1/G'(G^{-1}(v)) = q_r/\phi_\pi(G^{-1}(v))$, where $\phi(\cdot)$ denotes the PDF for $\mathcal{N}(\mu_\pi, \sigma_\pi^2)$. Applying mean value theorem to $G^{-1}$ on the interval $[Y_i, Y_{i+1}]$, we get:

$$G^{-1}(Y_{i+1}) - G^{-1}(Y_i) = (G^{-1})'(t_i)(Y_{i+1} - Y_i) \ ,$$

which is equal to $D_i^X$ since $Y_i = G(X_i)$, for some $t_i \in (Y_i, Y_{i+1})$.

Since $G^{-1}(v) \in I_r$, for all $v \in [0, 1]$, and $\phi_\pi(\cdot)$ is minimized on $I_r$ at the boundary, we also have:

$$(G^{-1})'(v) \leq \frac{q_r}{\phi_\pi(\mu_\pi + r)} \ .$$

Hence,

$$\max_i D_i^X \leq \frac{q_r}{\phi_\pi(\mu_\pi + r)} \max_i D_i^Y \ .$$

Since $\{Y_i\}_{i=1}^n$ uniform, similar to Theorem 2, we have:

$$\mathbb{E}\left[\max_i D_i^Y\right] = \frac{1}{n+1}\sum_{k=1}^{n+1}\frac{1}{k} \ .$$

So, for $n$ many points falling in $I_r$, we have:

$$\mathbb{E}[\epsilon_{\text{safe}}(s) \mid N_r = n] \leq \frac{q_r}{\phi_\pi(\mu_\pi + r)} \cdot \frac{1}{n+1}\sum_{k=1}^{n+1}\frac{1}{k} \ .$$

We can also bound the Gaussian term as:

$$\frac{q_r}{\phi_\pi(\mu_\pi + r)} = \sigma_\pi\sqrt{2\pi}\cdot\exp(r^2/2\sigma_\pi^2)(2\Phi(r/\sigma_\pi)-1)$$
$$\leq \sigma_\pi\sqrt{2\pi}\cdot\exp(r^2/2\sigma_\pi^2)(2\phi(0)\cdot r/\sigma_\pi)$$
$$= 2r\cdot\exp(r^2/2\sigma_\pi^2) \ .$$

Then, we have:

$$\mathbb{E}[S_{c,r}(s,a) - S_{c,r}^*(s,a) \mid N_r = n] \leq 2r\exp(r^2/2\sigma_\pi^2)\frac{L_{\text{UCB}}+L}{n+1}\sum_{k=1}^{n+1}\frac{1}{k} = O\left(r\exp(r^2/2\sigma_\pi^2)\cdot\frac{\log n}{n}\right) \ ,$$

which yields the desired result for fixed $r > 0$ and $\sigma_\pi > 0$.

$\square$

# B  Complete USPC Algorithms

In this section, we show the USPC algorithms that were used in our experiments. Blue parts denote the changes made by USPC over the original algorithm.

## B.1  USPC-TD3Lag

TD3Lag (Ji et al., 2024) is a primal-dual safe RL extension to TD3 (Fujimoto et al., 2018). Instead of simply training the policy to maximize rewards, TD3Lag introduces a dual variable $\lambda$ then constructs a Lagrangian as the new training objective. After each policy update, $\lambda$ is also updated according to episodic mean cost $\bar{C}$ and constraint threshold $h$. For USPC, we simply train $\hat{S}_c$ alongside the critics then replace the cost critic in the Lagrangian definition with $\hat{S}_c$, as shown in Algorithm 2. The parts modified by USPC are shown in blue.

---

**Algorithm 2** USPC-TD3Lag training for one epoch

---

**Require:** Batch size $B$, particle size $K$, policy parameter $\theta_i$
1: Sample $B$ transitions from replay buffer
2: Update $Q_r$, $Q_c$ via Bellman backup
3: Update $\hat{S}_c$ with Alg. 1
4: Update policy $\pi_\theta$ with TD3 update (Fujimoto et al., 2018) using loss $\mathcal{L}_\pi = \frac{1}{1+\lambda}\frac{1}{B}\sum_{i=1}^{B}\left(-\hat{Q}_r(s_i, a_i) + \lambda\hat{S}_c(s_i, a_i)\right)$
5: Update dual variable $\lambda$ using loss $\mathcal{L}_\lambda = -\lambda(\bar{C} - h)$

---

## B.2  USPC-CVPO

CVPO (Liu et al., 2022b) is a variational safe RL algorithm that consists of an expectation step to get an optimal variational policy distribution for the constrained problem and updates the dual variables, and a maximization step that fits the parametrized policy to the variational distribution with supervised learning. Since the maximization step does not depend on cost critics given the optimal variational distribution, we substitute $\hat{S}_c$ for $Q_c$ in the expectation step, as shown in Alg. 3. The parts modified by USPC are shown in blue.

---

**Algorithm 3** USPC-CVPO training for one epoch

---

**Require:** Batch size $B$, particle size $K$, policy parameter $\theta_i$
1: Sample $B$ transitions from replay buffer
2: Update $Q_r$, $Q_c$ via Bellman backup
3: Update $\hat{S}_c$ with Alg. 1
4: **for** $b = 1, \ldots, B$ **do**
5:     Sample $K$ actions $\{a_1, \ldots, a_K\}$ for $s_b$
6:     Compute $\{Q_r(s_b, a_k), \hat{S}_c(s_b, a_k); k = 1, ..., K\}$
7: **end for**
8: Compute optimal dual variables $\eta^*, \lambda^*$ by solving the convex optimization problem (9) in Liu et al. (2022b), substituting $\hat{S}_c$ in place of $Q_c$ in the original algorithm
9: Compute the optimal variational distribution for each state $\{q^*(\cdot|s_b); b = 1, ..., B\}$ by (8) in Liu et al. (2022b), substituting $\hat{S}_c$ in place of $Q_c$ in the original algorithm
10: Update policy from $\pi_{\theta_i}$ to $\pi_{\theta_{i+1}}$ via supervised learning objective (12) in Liu et al. (2022b)

---

## C   Comparison with COX-Q

In this section, we provide a more detailed comparison between USPC-CVPO and COX-Q (Li et al., 2026), a recent model-free safe RL baseline that is closely related to our setting. COX-Q follows a distributional RL approach for safety, in which they use an ensemble (of 5 networks) of cost critics, but instead of the UCB, they utilize quantile information.

**MDP setup.** In the COX-Q paper (Li et al., 2026), the Safety Gymnasium tasks are modified by reducing the episode length (steps) from 1000 to 400 while increasing the simulation time between two steps by $2.5\times$, which preserves the reward scale but rescales the cost by $0.4\times$, hence the cost thresholds are reduced accordingly. We report results under this setup in Figures 11a-12a and our setup in Figures 11b-12b for completeness. Unlike COX-Q however, we run our training in randomized environment layouts under both setups, which is a more challenging setting.

**Training performance comparison.** Across the full benchmark (Figures 3-5, Table 1), the comparison is task-dependent rather than uniformly favoring either method. In the final training averages reported in Table 1, USPC-CVPO satisfies the cost threshold on all six tasks. In contrast, COX-Q satisfies the threshold on Circle and HalfCheetahVelocity tasks, but exceeds it on Goal and Button tasks. On PointGoal and PointButton, USPC-CVPO improves both reward and cost relative to COX-Q, suggesting that safety propagation is especially useful in sparse-cost navigation tasks where local cost observations must generalize across nearby actions. On CarGoal, USPC-CVPO achieves substantially lower cost but also lower reward, indication a more conservative feasible policy. COX-Q is stronger on PointCircle, where it achieves both lower cost and slightly higher reward, and on HalfCheetahVelocity, where it obtains much higher reward while remaining below the cost threshold. On CarCircle, both methods are safe: USPC-CVPO obtains higher reward, while COX-Q obtains lower cost. Finally, USPC-CVPO appears to be more robust than COX-Q across different MDP setups for most tasks.

**Evaluation performance comparison.** Figure 12 shows that a similar pattern appears during evaluation, however evaluation costs can differ from training costs because evaluation uses the deterministic mean action whereas training includes stochastic exploration. USPC-CVPO is favorable on Goal and Button tasks, where COX-Q often remains above the cost threshold. COX-Q remains competitive on Circle tasks and is especially strong on HalfCheetahVelocity task. These results are consistent with the main experiments because USPC provides more consistent safety across the sparse-cost navigation tasks, while COX-Q can achieve better reward when its distributional cost estimates identify a high-reward feasible behavior.

**Interpretation.** The difference between the two methods is consistent with their design choices. COX-Q uses distributional cost information to guide exploration, which is very effective when the learned cost distribution is well calibrated and feasible high-reward region can be identified reliably. USPC instead constructs an epistemic upper confidence bound and then trains the SSN to propagate safety information across actions, which can make USPC more conservative but also more robust in tasks where safety hazards are sparse and cost critic is unreliable in early training. The HalfCheetahVelocity task results highlight a limitation of USPC, which is that in higher dimensional action spaces with tightly coupled reward and cost signals, a fixed anchor budget gives sparser action-space coverage, and the advantage of safety propagation is reduced.

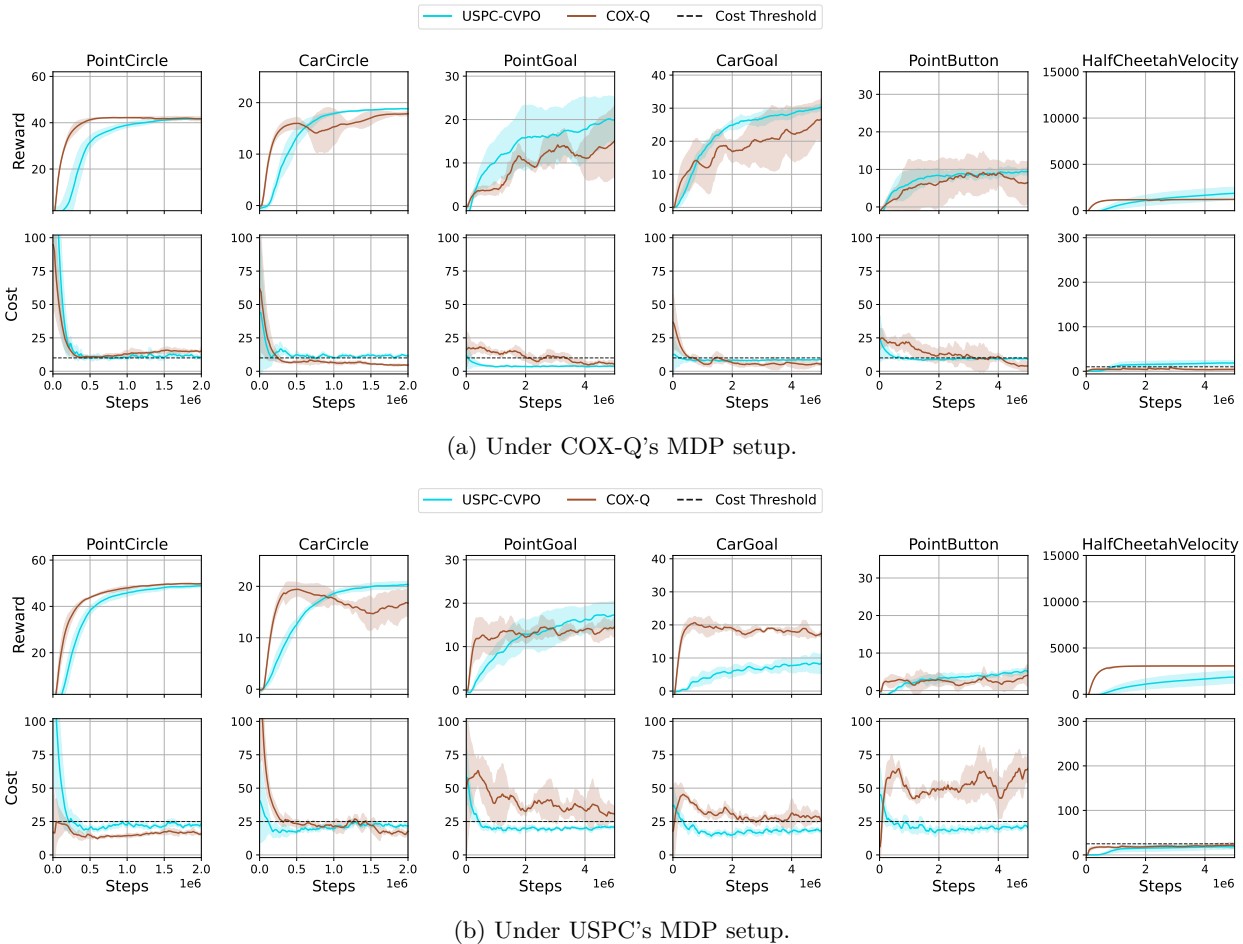

(a) Under COX-Q's MDP setup.

(b) Under USPC's MDP setup.

Figure 11: Comparison of USPC-CVPO with COX-Q across all six tasks on training episodes, under (a) the MDP setup used in the COX-Q paper and (b) our MDP setup. In the COX-Q paper (Li et al., 2026), the Safety Gymnasium tasks are modified by reducing the episode length (steps) from 1000 to 400 while increasing the simulation time between two steps by $2.5\times$, which preserves the reward scale but rescales costs by $0.4\times$, hence the cost thresholds are reduced accordingly (e.g., from 25 to 10 for PointGoal). Unlike COX-Q, however, we run our training in randomized environment layouts under both setups.

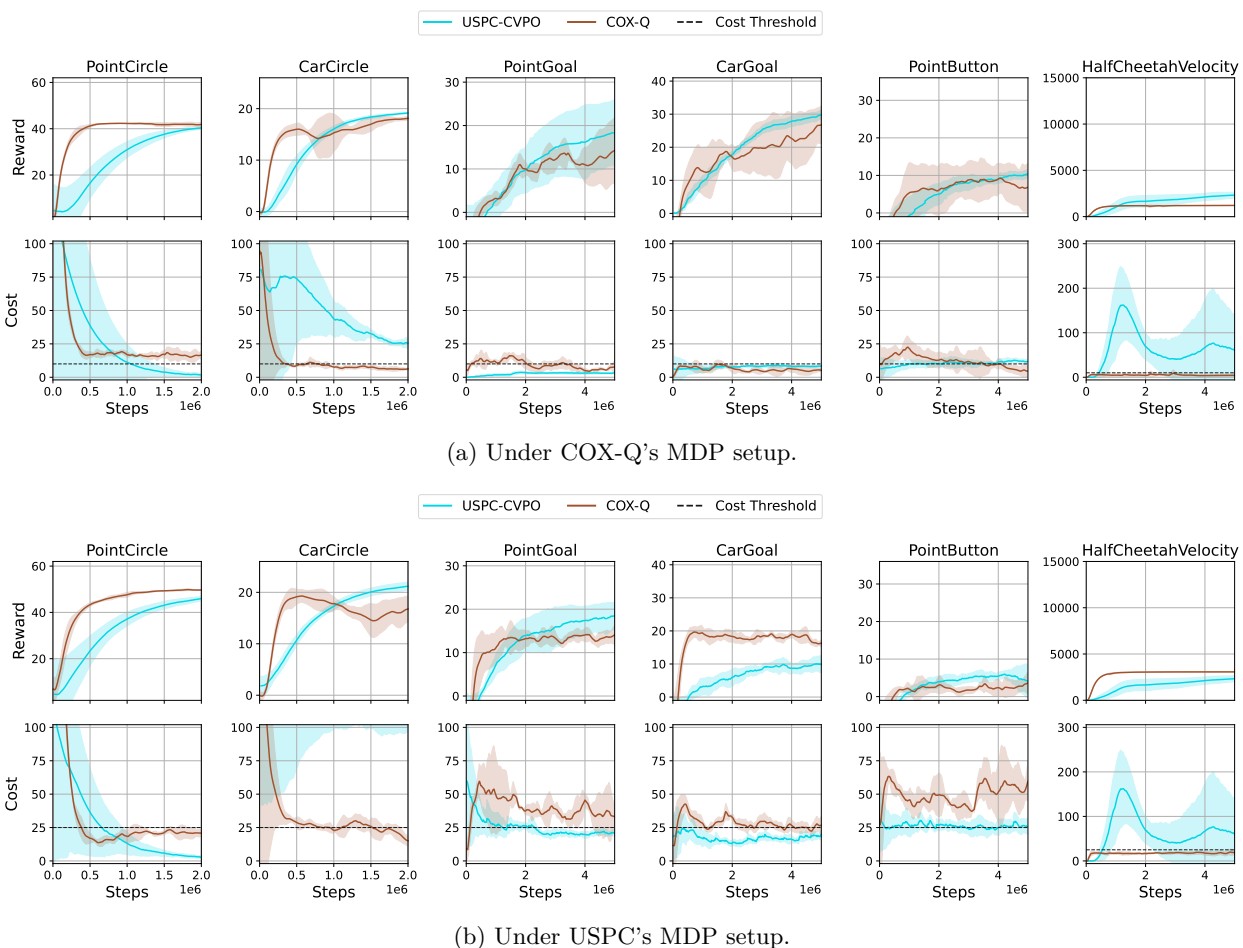

Figure 12: Comparison of USPC-CVPO with COX-Q across all six tasks on evaluation episodes, under (a) the MDP setup used in the COX-Q paper and (b) our MDP setup. In the COX-Q paper (Li et al., 2026), the Safety Gymnasium tasks are modified by reducing the episode length (steps) from 1000 to 400 while increasing the simulation time between two steps by 2.5×, which preserves the reward scale but rescales costs by 0.4×, hence the cost thresholds are reduced accordingly (e.g., from 25 to 10 for PointGoal). Unlike COX-Q, however, we run our training in randomized environment layouts under both setups.

## D   Hyperparameters

| Hyperparameter | Value | Description |
|---|---|---|
| $M$ | 6 | Number of cost critics in the ensemble |
| $\beta$ | 2.0 | Uncertainty scaling factor for UCB construction |
| $L$ | 2.0 (other tasks) 10.0 (Velocity) | Lipschitz constant assumed for $\hat{Q}_c$ |
| $K_{\mathrm{loc}}$ | 32 | Number of local anchor actions per batch sample |
| $K_{\mathrm{glob}}$ | 32 | Number of global anchor actions per batch sample |
| $\alpha$ | 0.005 | EMA/Polyak update rate for the SSN target network |
| $\mathrm{lr}_{\mathrm{SSN}}$ | $3 \times 10^{-4}$ | Learning rate of the safe set network |
| batch size | 256 | Batch size used for SSN updates |

Table 2: USPC-specific hyperparameters.

The values we use for the hyperparameters introduced by USPC are shown in Table 2. For other hyperparameters, we use the values provided by OmniSafe (Ji et al., 2024) (for TD3, CPO, TD3Lag and USPC-TD3Lag) and FSRL (Liu et al., 2024) (for CVPO and USPC-CVPO), and COX-Q (Li et al., 2026).

## E   Complete Experiment Results

We present full experiment results in this section.

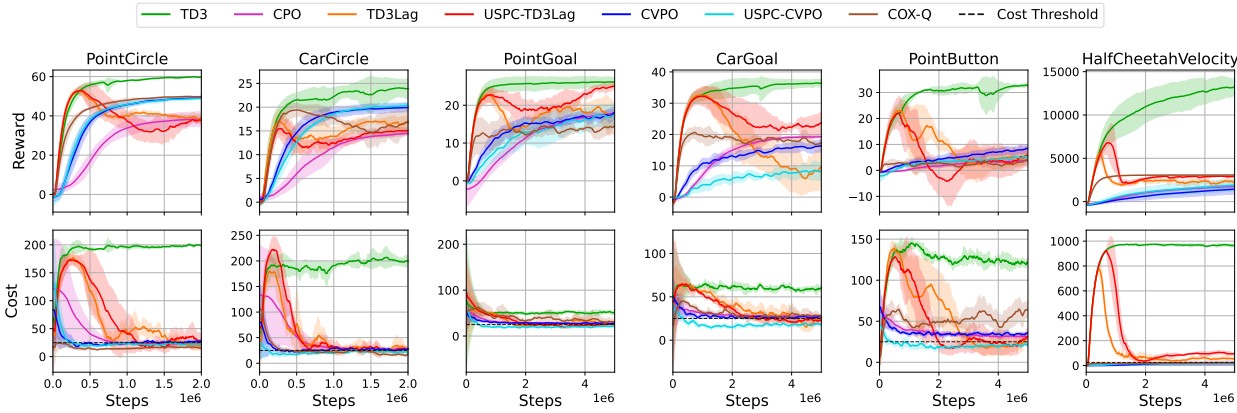

Figure 13: Reward and cost curves in training episodes, with y-axis unscaled.

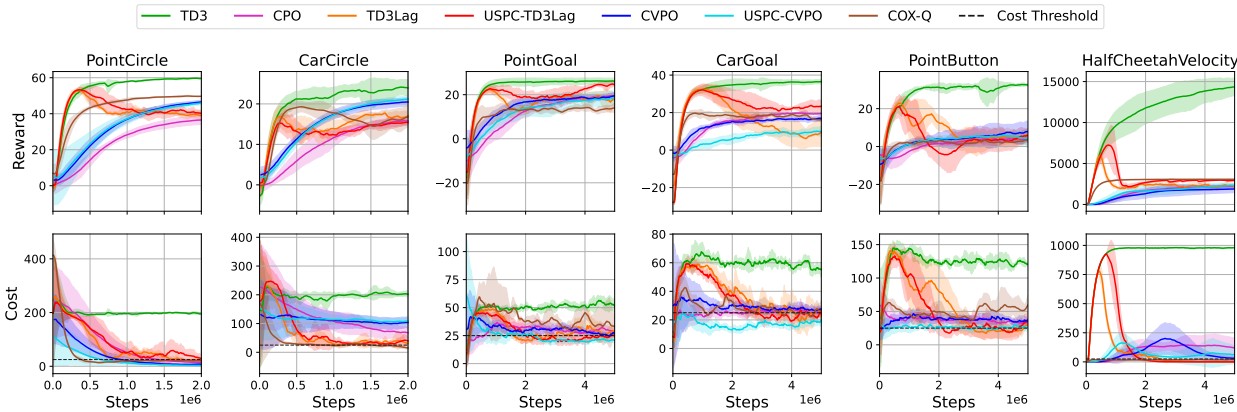

Figure 14: Reward and cost curves in evaluation episodes, with y-axis unscaled.

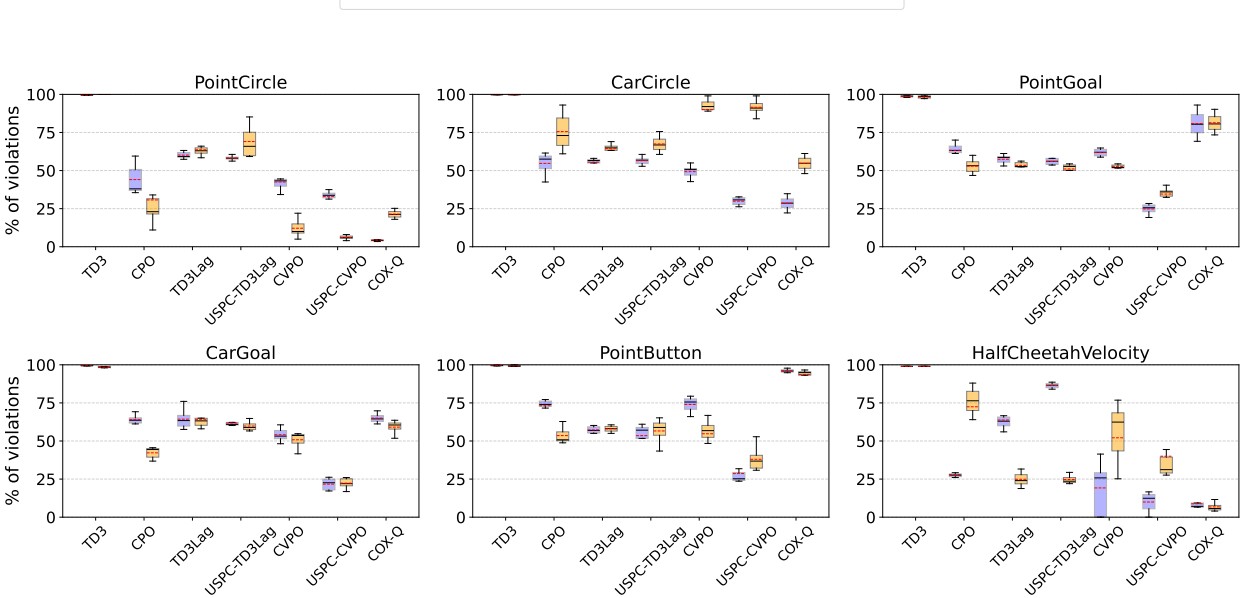

Figure 15: Constraint violation rates (including TD3), with y-axis scaled.

