# OpenReview forum: "Uncertainty-Aware Safety Propagation Critics for Safe Reinforcement Learning"
_TMLR — Accepted by TMLR_

### Review · Reviewer_wXin · 2026-03-30

**Summary Of Contributions:**

## Contributions
This paper proposes USPC (Uncertainty-Aware Safety Propagation Critics), which replaces the standard cost critic in safe reinforcement learning with a pessimistic surrogate. The surrogate is constructed from ensemble-based epistemic uncertainty, a UCB-style upper bound on cost, and Lipschitz-based safety propagation. The method is designed as a plug-in component for existing safe RL algorithms.

## Key strengths
The paper addresses a genuine bottleneck in safe RL: cost critics can be unreliable in under-explored regions, which can lead to constraint violations. The paper also makes an interesting conceptual connection between safe RL and safe Bayesian optimization by importing uncertainty-aware safety propagation ideas into the critic-learning setting. In addition, the proposed method is presented as plug-and-play for existing algorithms such as TD3-Lag and CVPO, which improves its practical relevance. Finally, the authors are appropriately candid about some of the limitations of their approach, including the reward-safety tradeoff and the increased computational cost caused by the ensemble and the additional safe set network.

## Key weaknesses
The degree of conceptual novelty relative to prior uncertainty-aware safe RL methods appears limited. The method feels more like a well-engineered combination of existing ideas than a fundamentally new approach. At the same time, the method is largely heuristic rather than theoretically grounded, yet the individual contribution of each design choice is not carefully isolated or studied. Methodologically, the SSN component is difficult to follow and not fully convincing, and it seems to introduce additional complexity as well as potential scaling issues in more complex environments. The empirical evidence is also not especially strong: experiments are limited to a small set of relatively simple environments, and the reported gains are modest, uneven, or absent in several cases. Finally, the discussion of prior work is not deep enough, and comparisons against stronger uncertainty-aware or risk-aware safe RL baselines are missing.

**Audience:**

Yes

**Audience Explanation:**

Yes, to some extent. The problem setting is important and the high-level idea is relevant to the TMLR audience. However, while the topic is interesting, I do not find the current evidence sufficiently convincing, so my concern is not about audience interest but about whether the paper’s findings are strong enough to warrant publication in its current form.

**Broader Impact Concerns:**

This work focuses on uncertainties on cost critic for safe reinforcement learning in simulated environments. Therefore, I don't have concerns on the ethical implications of the work.

**Claims And Evidence:**

No

**Claims Explanation:**

Although handling epistemic uncertainty is an important and well-motivated problem in safe RL, I do not think the current paper provides sufficiently strong evidence to support its main claims.

1.	Novelty appears limited.
The proposed method mainly combines an ensemble-based cost critic with a Safe Set Network (SSN) inspired by safety propagation in safe Bayesian optimization. While the combination is reasonable, the conceptual novelty seems incremental rather than substantial.

2.	The SSN / anchor-set mechanism is difficult to follow and not fully convincing.
In particular, the SSN target is generated by first filtering anchor actions using the current target SSN, and then propagating UCB-based costs across actions through a hand-tuned Lipschitz constant. This makes the method depend on several interacting learned approximations and hyperparameters, and it is not clear how reliable the resulting surrogate is in practice. I also did not fully understand the motivation for introducing the SSN at all. The paper states that the ensemble itself is not used directly for safety enforcement, and instead a separate SSN is trained to approximate the safety of state-action pairs. However, the reason this extra level of indirection is necessary is not convincingly justified. It also introduces substantial additional complexity and tuning burden, including the Lipschitz constant, anchor-set construction, anchor-set size, and an additional network to train.

3.	The anchor-set sampling procedure seems brittle and may not scale well.
Since the method relies on sampling a finite anchor set to approximate safety propagation in continuous action spaces, I am concerned about robustness and scalability, especially as action dimensionality grows. The paper lacks ablations on anchor-set size and composition, and more importantly, lacks analysis of how the method scales to larger action spaces. The HalfCheetahVelocity results make this concern more serious. In that environment, USPC does not show clear improvement over the baseline; for CVPO the learning appears unstable and the gains are unclear, while for TD3Lag the training result appears worse. This is plausible because HalfCheetahVelocity is likely harder for USPC than the other environments: it involves higher-dimensional control, stronger action-to-dynamics coupling, and a safety constraint defined through velocity rather than local obstacle geometry. The anchor-based action-space propagation used by USPC seems less well matched to this type of setting.

4.	The empirical evidence is not strong enough to support the paper’s claims.
The paper claims that USPC makes policies “less likely to violate cost constraints” and “consistently reduces both the frequency and magnitude of constraint violations while maintaining competitive reward performance.” I do not think the presented results support such a strong statement. The benefits appear uneven across environments and algorithms. As far as I can tell, only PointGoal clearly shows improved reward and cost in both training and evaluation. In several other cases, lower cost comes together with noticeably lower reward, which makes the result look more like a standard reward-safety tradeoff than a clear advantage of the proposed method. In principle, such a tradeoff can often be tuned in baseline methods as well. Moreover, given that the stated motivation is to better handle epistemic uncertainty, I would expect clearer improvements at evaluation time, but Figure 4 does not show obvious, consistent gains in evaluation performance.

5.	Comparison to stronger uncertainty-aware or risk-aware baselines is missing.
The discussion of related work and empirical comparisons should be strengthened. In particular, there are more relevant uncertainty-aware safe RL baselines that should be compared against, or at least discussed in more depth. For example, work by Stachowicz and Levine, and by Günster et al., also leverages ensemble-based uncertainty to improve safety, yet here these methods are only mentioned briefly under “conservative constraint enforcement.” Since epistemic uncertainty is central to the paper’s motivation, stronger comparisons along this axis seem necessary.

Additionally, there are quite some minor typos or mistakes. I just list some examples here:
1. Figure 5 is very hard to interpret, and I was unsure whether the legend entries for local anchor and global anchor may have been swapped.
2. After equation 12, the text mentions "$\alpha \in [0 ,1]$", but where is $\alpha$, I don't see it in Equation 12. This seems to be a typo or misplaced explanation.

**Requested Changes:**

### Critical issues (required for acceptance):
1.	Clarify and better justify the SSN and anchor-based design.
The Safe Set Network (SSN) and anchor-set-based safety propagation are difficult to follow and not fully convincing. In particular, the target construction depends on filtering anchors using the current SSN and propagating UCB-based costs via a hand-tuned Lipschitz constant. The authors should provide a clearer conceptual motivation for why this design is necessary (instead of directly using the ensemble/UCB critic), as well as a more transparent explanation of how the method works. A simplified formulation or ablation removing the SSN would help isolate its contribution.

2.	Provide ablations on key design components and hyperparameters.
The method depends on several interacting components (ensemble size, UCB scaling, Lipschitz constant L, anchor-set size and sampling strategy, SSN architecture). The paper should include systematic ablations to understand:
	- the impact of each component (e.g., UCB only vs. UCB + propagation vs. full method),
	- sensitivity to important hyper-parameters such as L (although it is shortly described, but Table 1 also shows that the Velocity env need a much larger L than the others) and anchor-set size,
	- the necessity of both local and global anchors.
This is critical to assess robustness and reproducibility.

3.	Strengthen empirical evaluation and claims.
The current results do not convincingly support the claim of consistent improvement. The authors should:
- moderate claims such as “consistently reduces violations while maintaining competitive reward”, and
- provide stronger empirical evidence (e.g., additional runs, clearer statistics, or improved performance).
In particular, the inconsistent behavior on HalfCheetahVelocity should be analyzed in more depth, rather than only briefly discussed.

4. Add comparisons with stronger uncertainty-aware or risk-aware baselines.
Since the main motivation is handling epistemic uncertainty, comparisons should include more relevant baselines (or at least detailed discussion), such as recent uncertainty-aware safe RL or risk-sensitive methods (e.g., ensemble-based uncertainty approaches, CVaR-style methods, or recent works by Stachowicz et al. and Günster et al.). This is important to properly position the contribution.

5.	Discuss scalability and limitations more rigorously.
The anchor-based sampling procedure may not scale well to higher-dimensional action spaces. The authors should provide either empirical evidence or analysis of how performance depends on action dimensionality and anchor-set size. The weak performance on HalfCheetahVelocity suggests potential limitations that should be explicitly addressed.

6.	Provide deeper analysis of reward–safety tradeoff.
Since the method often reduces cost at the expense of reward, a more explicit analysis (e.g., Pareto-style comparison or tuning curves), compared with baselines, would help clarify whether improvements go beyond standard tradeoffs.

### Additional suggestions (would strengthen the paper):

1.	Improve clarity of presentation.
- Simplify or better explain the SSN formulation and training procedure.
- Provide intuition or diagrams illustrating anchor selection and safety propagation.
2.	Improve figures and diagnostics.
- Figure 5 is difficult to interpret; the legend (local vs. global anchors) should be checked and clarified.
- Add clearer quantitative summaries (e.g., mean ± std over seeds) to support claims.
3.	Address minor issues and typos.
- After Equation 12, the mention of $\alpha \in [0,1]$ appears inconsistent, as $\alpha$ is not defined in that equation.
- Broken references to table in Appendix B.
- Carefully proofread for similar minor inconsistencies.

---

> ### Author Response · Authors · 2026-04-14
> **Response to Reviewer wXin**
>
> We thank the reviewer for their thorough review and valuable suggestions.
>
> > 1. Clarify and better justify the SSN and anchor-based design. The Safe Set Network (SSN) and anchor-set-based safety propagation are difficult to follow and not fully convincing. In particular, the target construction depends on filtering anchors using the current SSN and propagating UCB-based costs via a hand-tuned Lipschitz constant. The authors should provide a clearer conceptual motivation for why this design is necessary (instead of directly using the ensemble/UCB critic), as well as a more transparent explanation of how the method works. A simplified formulation or ablation removing the SSN would help isolate its contribution.
>
> We agree that in the submitted version of the manuscript, the motivation is not entirely clear. We have added a paragraph to our Section 4.2 to justify how our method helps at a conceptual level. We also include the same explanation below for your convenience:
>
> "A natural question to ask is why the ensemble UCB itself cannot simply replace the cost critic. The UCB provides a pointwise pessimistic cost estimate: at each (s, a) it inflates the predicted cost based on local ensemble disagreement. However, this treats every action independently. In continuous action spaces, policies often select actions in regions with sparse data coverage, where the UCB may be overly conservative near well-explored safe actions, or insufficiently conservative in unexplored regions. The SSN addresses this by encoding a structural relationship: the safety status of an action depends not only on its own UCB, but on whether nearby actions are known to be safe. In other words, the SSN allows us to propagate safety across different actions."
>
> In addition to the above high-level explanation, we have also added theoretical and empirical results to further support our claims. First, we have added Theorem 1 to show that the SSN provides a more conservative estimate than the ensemble mean, while at the same time it is less conservative than using UCB directly for state-action pairs that are deemed safe (and thus exploration is not hindered as much in safe regions). To empirically support our claims, we have added an ablation study where we compare the full USPC algorithm with simply replacing the cost critic with an ensemble UCB. The results show that while this UCB variant achieves similar cost, the reward is much lower, which also suggests that directly using UCB hinders exploration.
>
> > 2. Provide ablations on key design components and hyperparameters. The method depends on several interacting components (ensemble size, UCB scaling, Lipschitz constant L, anchor-set size and sampling strategy, SSN architecture). The paper should include systematic ablations to understand: the impact of each component (e.g., UCB only vs. UCB + propagation vs. full method), sensitivity to important hyper-parameters such as L (although it is shortly described, but Table 1 also shows that the Velocity env need a much larger L than the others) and anchor-set size, the necessity of both local and global anchors. This is critical to assess robustness and reproducibility.
>
> We have added new ablation studies for the ensemble size, anchor set size, anchor distribution (global vs. local), and as mentioned above, the "UCB-only vs. USPC" ablation. We hope that these address your concerns.
>
> You are also right to question why the HalfCheetahVelocity task requires a different value of L. The intuitive reason is that by design, it is possible in the Velocity environments to incur much more cost compared to the other tasks (TD3 incurs ~900 episodic cost) since the costs are given for being above a certain velocity (which is directly encouraged by the reward signal).
>
> Regarding the necessity of local and global samples: Ideally, given enough compute, we would like to have a large number of global samples (or perhaps even a fine grid) to sufficiently cover the action space. However, scalability can be a problem with sampling-based methods (as you also mentioned). In large (both in terms of dimension and the range of values) action spaces, local anchors aim to provide a more stable set of points for SSN target computation at the cost of "exploration" for safety propagation.
>
> (continued)

---

> ### Author Response · Authors · 2026-04-14
> **Response to Reviewer wXin (part 2)**
>
> > 4. Add comparisons with stronger uncertainty-aware or risk-aware baselines. Since the main motivation is handling epistemic uncertainty, comparisons should include more relevant baselines (or at least detailed discussion), such as recent uncertainty-aware safe RL or risk-sensitive methods (e.g., ensemble-based uncertainty approaches, CVaR-style methods, or recent works by Stachowicz et al. and Günster et al.). This is important to properly position the contribution.
>
> We have added COX-Q, which has recently been published at ICLR 2026, as a new baseline. Due to time constraints the results are currently only in PointGoal; we will be getting results for the rest of the tasks in the coming days. We hope that this result is informative.
>
> Regarding why we did not include Stachowicz and Günster papers as empirical baselines: Stachowicz & Levine (RACER) does not operate within the CMDP framework. Safety violations are encoded as episode terminations with zero return rather than through a separate cost function with a cumulative cost threshold. There is no cost critic or threshold, the components USPC is designed to augment. RACER was also evaluated on custom driving environments rather than Safety Gymnasium, so a fair empirical comparison on our benchmarks would require substantially redesigning one of the two methods. Günster et al. (D-ATACOM) requires a known dynamics model to construct the constraint manifold (making it partially model-based), enforces stepwise constraints rather than cumulative cost thresholds, and was evaluated on custom robotics environments (Cartpole, Navigation, Air Hockey) that differ from Safety Gymnasium. These differences in problem formulation, constraint type, and evaluation domain make direct empirical comparison difficult to set up fairly. We chose COX-Q as our additional baseline because it is a recent method that shares the CMDP formulation, benchmark suite, and ensemble-based design with USPC, which hopefully allows for a more meaningful comparison.
>
> > 5. Discuss scalability and limitations more rigorously. The anchor-based sampling procedure may not scale well to higher-dimensional action spaces. The authors should provide either empirical evidence or analysis of how performance depends on action dimensionality and anchor-set size. The weak performance on HalfCheetahVelocity suggests potential limitations that should be explicitly addressed.
>
> We understand your concern about scalability. While it was not previously discussed in detail, scalability is a problem many sampling-based methods face, in the sense that it is difficult to sufficiently cover a high-dimensional continuous space using a finite number of samples. We now added a new theoretical result (Theorem 2) to more rigorously address the optimality gap arising from using the anchor set approximation, as well as the space coverage of the anchor set. We have also updated our limitations accordingly.
>
> > 6. Provide deeper analysis of reward–safety tradeoff. Since the method often reduces cost at the expense of reward, a more explicit analysis (e.g., Pareto-style comparison or tuning curves), compared with baselines, would help clarify whether improvements go beyond standard tradeoffs.
>
> We agree that such a comparison would be informative. We have added a Pareto frontier plot, which shows that for most tasks, USPC results dominate the base algorithms, which suggests that USPC provides improvements beyond simply trading off reward for lower costs. We believe that this is possible because SSN can essentially allow for "selective conservativeness" (please see Section 4.2.1 in our revised manuscript).
>
> > Minor errors.
>
> We have fixed the errors you have mentioned; thank you for bringing them to our attention.

---

### Review · Reviewer_ZDpQ · 2026-03-31

**Summary Of Contributions:**

The paper proposes a new algorithm which combines anchor set generation, action-value extrapolation using first-order approximation under Lipshitz assumptions, and accounts for epistemic uncertainty through ensembles of cost critics and a UCB criterion. The method aims to improve the safety constraints of cost critics. The paper shows that this technique is quite generalisable since it is applicable to different baselearners, as demonstrated on TD3Lag and CVPO. The algorithm seems to be well-constructed and have good empirical performance. However, the novelty and efficiency of the individual components, the theoretical backing, and comparison to algorithms that try to do the same thing (epistemic uncertainty + constraints) seems to be limited. Moreover, the scope of epistemic uncertainty seems to be limited since the only uncertainty seems to be in the initialisation of Q-models.

**Audience:**

Yes

**Audience Explanation:**

Safe model-free RL is a topic of strong interest to the machine learning community. While the technique may not have application to model or reward uncertainty, it is of general interest to provide a solid way to reduce the cost violations. The technique applies to a variety of baselearners which makes it more applicable.

**Broader Impact Concerns:**

No ethical implications.

**Claims And Evidence:**

No

**Claims Explanation:**

- The chosen domains and baselearners seem reasonable.

- There is not much theory development.

- There is also no baseline comparison beyond the comparison to the baselearner; it would be insightful and help to see how the USPC would compare to other means of robustifying the cost critic. The work uses a lot of tricks that are similar to prior works but does not compare to the original methods.

- Since the USPC part consists of several components, it would be insightful to see in some way why all the different components are needed and how efficient they are in achieving their purpose, either through theory or via an ablation study. E.g. what happens when some of your components are replaced by alternative choices.

- Assumptions. the L-lipschitz assumption on the Q-value as a function of the action space is also a significant assumption

- The related work discussion is at times extensive and insightful. However, the review of safe model-free RL is limited to before 2020. Moreover, the discussion of techniques combining safe model-free RL and epistemic uncertainty is especially limited.

- The definition of epistemic uncertainty in this paper seems to be limited to the cost critic estimates obtained from learning under a different starting initialisation while otherwise following the same exact learning routine on the same data etc. Perhaps some caveat should be made about the limits of the application of the technique to more challenging settings. For instance, I would not expect the present technique to work well for uncertain environment models or reward functions.

**Requested Changes:**

To summarise the key points I made earlier, these points would improve the paper:

- theoretical development and clarification of further assumptions
- comparisons to other means of incorporating epistemic uncertainty into the cost critic
- comparisons to other means of achieving the individual components of the USPC
- addressing how your technique positions into more general forms of uncertainty (e.g. in rewards, transition dynamics)
- updated literature review on safe model-free RL and its intersection with epistemic uncertainty approaches

---

> ### Author Response · Authors · 2026-04-14
> **Response to Reviewer ZDpQ**
>
> We thank the reviewer for their detailed and insightful comments.
>
> > There is not much theory development.
>
> We understand your concern. To more rigorously assess USPC, we have added several theoretical results:
> - Conservativeness of SSN compared to cost critic ensemble mean (Theorem 1) and true cost Q-value (Corollary 1)
> - Lipschitzness of the UCB (Lemma 1)
> - Optimality gap arising from the use of a finite anchor set compared to the true continuous space (Theorem 2 and Proposition 1)
>
> We hope that these additions provide a better understanding of our method.
>
> > There is also no baseline comparison beyond the comparison to the baselearner; it would be insightful and help to see how the USPC would compare to other means of robustifying the cost critic. The work uses a lot of tricks that are similar to prior works but does not compare to the original methods.
>
> We have added COX-Q (Li et al., 2026) as an additional baseline. Due to time constraints the results are currently only in PointGoal; we will be getting results for the rest of the tasks in the coming days. COX-Q also uses a deep ensemble of cost critics but takes a distributional approach to quantify and use uncertainty, and was recently published at ICLR 2026. Importantly, COX-Q operates within the same CMDP framework and Safety Gymnasium benchmarks as USPC, enabling a direct and fair methodological comparison. We have also expanded the related work to discuss the two works the reviewer mentioned in greater depth.
>
> > Since the USPC part consists of several components, it would be insightful to see in some way why all the different components are needed and how efficient they are in achieving their purpose, either through theory or via an ablation study. E.g. what happens when some of your components are replaced by alternative choices.
>
> Although we had tried to address this concern partly with the L and β ablations and our discussions, we agree that it has room for improvement. Thus, we have made a number of additions that are described below.
>
> First, we added ablation studies for ensemble size, anchor set size and distribution, and the propagation mechanism itself, i.e. replacing the cost critic with an ensemble UCB instead of SSN. We hope these provide some insight about these components.
>
> Note that, as discussed in the paper, we use an anchor set-based approximation as opposed to full inner loop optimization for practical purposes (computational simplicity and cost), although we have now also provided theoretical results for the anchor set's coverage of the action space and the optimality gap arising from its use (compared to optimization in the continuous space) to more rigorously assess the tradeoff.
>
> We have also added the "Using a safe set net to estimate the safe set." section to our discussion to explain our reasons for using a safe set network (SSN) as opposed to directly using the targets that the SSN uses. In summary, the reasons are analogous to why critics are used in RL: the use of SSN yields lower-variance values and gradients as well as better generalizability.
>
> > Assumptions. The L-Lipschitz assumption on the Q-value as a function of the action space is also a significant assumption.
>
> You are correct that this is not a trivial assumption in theory. As discussed in the "Lipschitz assumptions." section of our discussion, it is indeed possible for underlying Q-functions to be discontinuous (e.g. in a sparse reward or cost setting), but since neural networks (which learn continuous functions by design) are used as an estimator, we believe Lipschitzness is a fair assumption to make. As for the value of L, we chose to treat it as a hyperparameter for simplicity; in theory it is possible to combine USPC with methods such as (Virmaux & Scaman, 2018; Bjorck et al., 2021) to obtain a more rigorous value for L, if desired.
>
> > The related work discussion is at times extensive and insightful. However, the review of safe model-free RL is limited to before 2020. Moreover, the discussion of techniques combining safe model-free RL and epistemic uncertainty is especially limited.
>
> We appreciate your concern about the coverage of our related work. We have expanded our related work to include more recent papers and topics.
>
> (continued)

---

> ### Author Response · Authors · 2026-04-14
> **Response to Reviewer ZDpQ (part 2)**
>
> > The definition of epistemic uncertainty in this paper seems to be limited to the cost critic estimates obtained from learning under a different starting initialisation while otherwise following the same exact learning routine on the same data etc. Perhaps some caveat should be made about the limits of the application of the technique to more challenging settings. For instance, I would not expect the present technique to work well for uncertain environment models or reward functions.
>
> We thank the reviewer for this point. USPC is a model-free method, so transition and reward uncertainty are experienced through sampling rather than learned models. The relevant epistemic uncertainty is in the cost critic, which is exactly what our ensemble targets, analogous to how model-based methods apply ensembles to dynamics models. Extending USPC to address aleatoric uncertainty explicitly can be an interesting research direction.
>
> We discuss ensemble limitations in Section 6. We agree that initialization-only diversity can underestimate uncertainty when members converge similarly. A practical mitigation is training ensemble members on different data subsets (bootstrapping), which increases diversity at the cost of weakening the i.i.d. assumption underlying our CLT argument (which we use to argue replacing the Gaussian process in previous safe BO methods with our ensemble). We have extended the discussion to address this tradeoff explicitly (in the "Approximating the uncertainty with neural networks" section).

---

### Review · Reviewer_omZF · 2026-04-01

**Summary Of Contributions:**

This work proposes Uncertainty-Aware Safety Propagation Critics as an algorithm for safe RL. They replace the cost critic with an ensemble of critics and a safe network that learns to approximate the safety.

Pros:
- Addressing an important area in the field of safe RL to move towards practical/lower compute/complexity safe RL algorithms (plus clear ways to tradeoff/give a lever for safety)
- Ports nicely into existing AC algos

Considerations:
- The training/eval difference is quite noticeable, should warrant more exploration than just the simple paragraph
- Further analysis of the epistemic uncertainty seems important, since this is a core argument of the paper but the num ensemble isn't ablated (and the calibration of the uncertainty isn't evaluated)

**Additional Comments:**

Questions:
- What happens if for the S^target none are less than h?

**Audience:**

Yes

**Audience Explanation:**

Safe RL is a field of broad interest, and this paper presents an algorithm in an interesting direction.

**Claims And Evidence:**

No

**Claims Explanation:**

In terms of support, it is close to a yes, but needs just a little more.

- It isn’t totally clear from figure 3/4 how statistically significant the results are. It would benefit from perhaps additional tables in the appendix, or even further analysis (e.g. IQM, that sort of stuff). Figure 5 is more in that direction, but would also benefit from comparing the reward axis. Perhaps putting Figure 5 before 3/4 and consolidating them (or making a table)
- The claims about L seem overstated. The overlap in confidence is significant, and it’s hard to draw meaningful claims. But the fact that L=0 performs quite well seems important?

**Requested Changes:**

Addressing the above points, and also these minor nits:
- Extra space after (USPC) in intro
- The arrow on the query action in Figure 1(c) doesn’t show up like the others
- Figure 3 and 4 the cost limit is different sides of the legend
- “This is” sentence ends in 5.3?
- Figure 5 has the cost threshold in red in the legend but not in the plot
- Perhaps there is some form of normalized reward/cost that would make for an interesting plot. E.g. normalize reward (according to max of the environment) and scale by cost to give some example of what the tradeoffs are (or perhaps even some sort of frontier plot of the two on x/y axis)
- There is a push task mentioned, but it isn’t in the figures?

---

> ### Author Response · Authors · 2026-04-14
> **Response to Reviewer omZF**
>
> We thank the reviewer for their constructive feedback and helpful suggestions.
>
> > The training/eval difference is quite noticeable, should warrant more exploration than just the simple paragraph ... It isn't totally clear from figure 3/4 how statistically significant the results are. It would benefit from perhaps additional tables in the appendix, or even further analysis (e.g. IQM, that sort of stuff). Figure 5 is more in that direction, but would also benefit from comparing the reward axis. Perhaps putting Figure 5 before 3/4 and consolidating them (or making a table)
>
> We understand your concern about the statistical significance. We had briefly commented on this difference in the "Training and evaluation cost discrepancy" part of our results section, but we have elaborated further in that regard in our updated manuscript. We also understand that although we had included the mean and confidence intervals in our reward/cost plots, it can be hard to distinguish the details with a human eye (especially with a printed copy), so we have added a table containing mean ±std reward/cost values (averaged over the last 50 episodes), which we hope clarifies our results.
>
> > Further analysis of the epistemic uncertainty seems important, since this is a core argument of the paper but the num ensemble isn't ablated (and the calibration of the uncertainty isn't evaluated)
>
> We acknowledge that a formal calibration analysis would be a valuable addition. However, we note that USPC does not require the ensemble uncertainty to be well-calibrated in the classical sense (i.e. correct frequentist coverage). Rather, our method only requires that ensemble disagreement is monotonically related to true epistemic uncertainty; that is, regions with less data coverage produce higher disagreement than well-explored regions. This ordering property is sufficient for our UCB construction to be effective: it ensures that the safe set network assigns higher predicted costs in poorly explored regions, regardless of the exact scale of the uncertainty.
>
> This property has been empirically demonstrated for point-estimate neural network ensembles in prior RL work. Osband et al. (2016) showed that bootstrapped neural networks with random initialization produce uncertainty estimates that visibly expand away from training data, even without learned per-member variance (similar to our setup). Lee et al. (2021, SUNRISE) also validate this by using point-estimate Q-network ensemble standard deviation to construct a UCB for exploration, relying on the same assumption that ensemble disagreement tracks epistemic uncertainty in Q-value estimates.
>
> Furthermore, the hyperparameter β can at least partially absorb any miscalibration by rescaling the raw ensemble standard deviation. Our β ablation (Figure 6b) confirms that varying β smoothly controls the safety-reward tradeoff, which should not be possible if the uncertainty estimates were uninformative. We believe this empirically supports that the ensemble disagreement is at least mostly well-ordered, even if not perfectly calibrated. Augmenting USPC with a more direct approach to calibration can be an interesting extension.
>
> We agree that ablating the number of ensemble members M is informative, and updated our manuscript with an ablation study. In our preliminary experiments, we found that M ≥ 3 produced stable uncertainty estimates, with diminishing returns beyond M = 6. We chose M = 6 as a practical default that balances uncertainty quality with computational overhead.
>
> > The claims about L seem overstated. The overlap in confidence is significant, and it's hard to draw meaningful claims. But the fact that L=0 performs quite well seems important?
>
> We understand your concern. We agree that our results imply the improvements of USPC mainly come from the overall propagation mechanism (i.e. learning a safety value that aggregates Q-value estimates across actions) rather than the Lipschitz distance term in particular (please see the comparison with UCB-CVPO in Figure 11 of our revised manuscript). We have revised the wording of our claims in that part. In previous safe Bayesian optimization works that have inspired USPC, a Lipschitz term (or similar) is used to allow the optimization to explore parts of the parameter space while never querying an unsafe point, which is not feasible in our model-free RL setting. Instead, in our work, the Lipschitz distance term acts as regularization to discourage SSN targets from selecting actions from wildly different parts of the action space, where the agent is more likely to be uncertain. In other words, it allows the selected action to be shifted more smoothly over the training, somewhat similarly to how the policy itself is trained.
>
> (continued)

---

> ### Author Response · Authors · 2026-04-14
> **Response to Reviewer omZF (part 2)**
>
> > Perhaps there is some form of normalized reward/cost that would make for an interesting plot. E.g. normalize reward (according to max of the environment) and scale by cost to give some example of what the tradeoffs are (or perhaps even some sort of frontier plot of the two on x/y axis)
>
> We agree with your feedback. Since different methods tend to yield different rewards and costs at the same time, it is hard to determine whether one method is better than the other from the reward/cost curves alone. To remedy this issue, we have added a Pareto plot, which shows that the empirical Pareto frontier for the reward-cost tradeoff is mostly defined by USPC runs. We hope that this result provides further insight.
>
> > What happens if for the S^{target} none are less than h?
>
> This is a valid point. In the case that all targets are unsafe, we simply use the UCB of the query point as our target, although in the paper we previously forgot to mention this detail, thank you for bringing this to our attention. We have since added a mention of this fallback mechanism to Section 4.2 and Algorithm 1.
>
> > Typos and minor errors.
>
> Thank you for your attention to detail. We have fixed the typos and errors you have mentioned.
>
> "The arrow on the query action in Figure 1(c) doesn't show up like the others": That is on purpose. Figure 1c shows that points on the UCB are lifted upwards by the Lipschitz distance term. Since the query point's distance to itself is zero, it is not lifted. We hope that this clarifies the figure.

---

### Author Response · Authors · 2026-04-14
**Summary of Revisions**

We thank the editor and the reviewers for their effort. In our responses to our reviewers, we addressed the concerns of each reviewer in detail. Below, we summarize their main concerns:

- **Lack of modern baselines.** In our initial manuscript, we aimed to examine the impact of our method on previous, more established methods such as TD3Lag and CVPO, as well as an unsafe baseline (TD3) and a foundational safe RL method (CPO). However, the reviewers (Reviewer ZDpQ and wXin) rightfully pointed out that our method should also be compared with state-of-the-art works that use a similar approach for safety. To that end, we have added comparisons with COX-Q, another uncertainty-aware model-free safe RL algorithm. Due to time constraints we were unable to get results for all tasks; we will be working on this in the following days.

- **Lack of empirical results and analysis.** Our initial manuscript contained reward and cost curves, constraint violation rates and ablation studies for the value of assumed Lipschitz constant and confidence interval coefficient. However, all three reviewers asked for new experiment results and analysis to improve the empirical grounding of our paper. To address their concerns, we added ablation studies for the anchor set size and distribution, ensemble size, the propagation mechanism itself (i.e. the USPC machinery vs. simply replacing the cost critic with an ensemble UCB), and a Pareto frontier plot to assess the reward-cost tradeoff. In addition, we have expanded the discussion in many places, including an extended motivation, calibration, a new section comparing the SSN approach with a Monte Carlo-like target, scalability limitations, explanation for the different behavior in HalfCheetahVelocity, extended discussion of training-evaluation discrepancy, role of the Lipschitz distance term, and more.

- **Insufficient theoretical results.** While our initial manuscript contained some high-level theoretical discussion and motivation, it did not contain any rigorous statements regarding the properties of USPC. Reviewers wXin and ZDpQ rightfully pointed out that such results would be beneficial for the paper. We have since added a number of theoretical discussion of results, including for conservativeness of the safe set network, the action space coverage of global and local anchors, and the expected optimality gap arising from the proposed anchor set approximation.

We believe that following the detailed feedback from our reviewers, we have substantially improved our manuscript and uploaded the revised manuscript (newly added/modified sections are highlighted in blue). We hope that our revisions adequately address the concerns raised during the review process.

---

### Decision · Action_Editor_ZSzu · 2026-05-12

**Recommendation:** Accept with minor revision

**Additional Comments:**

As stated above, the requested revision is to evalute COX-Q on the remaining environments.

**Audience:**

Yes

**Audience Explanation:**

I agree with all reviewers, that safe reinforcement learning is an important topic and that at least some individuals will be interested in the specific approach proposed in the submission

**Claims And Evidence:**

Yes

**Claims Explanation:**

While the claims were initially not sufficiently well supported, the revision added some theoretical support and additional experiments which convinced two reviewers. Reviewer wXin argues that the claims are not sufficiently supported, partially due to a perceived lack of novelty (which I agree is limited, but sufficient for TMLR's threshold), but also due to limited empirical support. I tend to somewhat agree with the later point, but argue that **after evaluating COX-Q on the other environments (as promised by the authors)**, the empirical support will be sufficient.

Hence, I recommend "accept with minor revision", where the additional experiments will be mandatory for acceptance (irrespective of their outcome).

---

> ### Author Response · Authors · 2026-06-12
>
> Dear Action Editor,
>
> We have uploaded our camera-ready revision.
>
> As requested, we evaluated COX-Q on additional tasks under both the original COX-Q MDP setup and our MDP setup. We have updated our existing figures to include the COX-Q results. In addition, we moved our previous COX-Q and USPC-CVPO comparison figure, together with an extended discussion, to Appendix C. We also reviewed the manuscript for typos, coherence, and other camera-ready requirements.
>
> Thank you for your diligent work in handling our submission and reviewing process.
>
> Best,
>
> The authors